# Anomalies in the Carbonate System of Red Sea Coastal Habitats

Kimberlee Baldry[1*], Vincent Saderne[1], Daniel C. McCorkle[2], James H. Churchill[2], Susana Agusti[1] and Carlos M. Duarte[1]

[1] Red Sea Research Center and Computational Bioscience Research Center, King Abdullah University of Science and Technology (KAUST), Thuwal, 23955, Saudi Arabia
[2] Woods Hole Oceanographic Institution (WHOI), Woods Hole, Massachusetts 02543, USA
[*] Current affiliation: Institute of Marine and Antarctic Studies, University of Tasmania, Hobart 7000, Australia

*Correspondence to*: Kimberlee Baldry (kimberlee.baldry@utas.edu.au)

**Abstract.** We use observations of dissolved inorganic carbon (DIC) and total alkalinity (TA) to assess the impact of ecosystem metabolic processes on coastal waters of the eastern Red Sea. A simple, single-end-member mixing model is used to account for the influence of mixing with offshore waters and evaporation/precipitation, and to model ecosystem-driven perturbations on the carbonate system chemistry of coral reefs, seagrass meadows and mangrove forests. We find that 1) along-shelf changes in TA and DIC exhibit strong linear relationships that are consistent with basin-scale net calcium carbonate precipitation; 2) ecosystem-driven changes in TA and DIC are larger than offshore variations in >70% of sampled seagrass meadows and mangrove forests, changes which are influenced by a combination of longer water residence times and community metabolic rates; and 3) the sampled mangrove forests show strong and consistent contributions from both organic respiration and other sedimentary processes (carbonate dissolution and secondary redox processes), while seagrass meadows display more variability in the relative contributions of photosynthesis and other sedimentary processes (carbonate precipitation and oxidative processes). The results of this study highlight the importance of resolving the influences of water residence times, mixing and upstream habitats on mediating the carbonate system and coastal air-sea carbon dioxide fluxes over coastal habitats in the Red Sea.

## 1.    Introduction

Coral reefs, seagrass meadows and mangrove forests are sites of intense metabolic processes.  These habitats are distributed heterogeneously in the coastal zone, at shallow depths where perturbations in the carbonate system by metabolic processes can have the greatest influence on water chemistry and air-sea carbon dioxide ($CO_2$) exchange (Bauer et al., 2013; Camp et al., 2016; Cyronak et al., 2018; Gattuso et al., 1998; Guannel et al., 2016; Unsworth et al., 2012).

The cumulative impact of coastal habitats on the carbonate system, along with their overall importance in the global carbon cycle, is difficult to quantify and is poorly represented when compared to knowledge of open ocean processes (IPCC, 2014). The open ocean is geographically separated from the benthos and land so their respective influences on the carbonate system often can be ignored over short time-scales. In addition to the influence of metabolism in coastal habitats, the carbonate system of the coastal zone is also influenced by both the benthos

and land over short time-scales. Thus, terrestrial and freshwater inputs (dissolved and particulate), sediment exchanges, biological processes, and changes in circulation and water residence time must all be considered when studying perturbations in the carbonate system of the coastal zone (Doney, 2010; Duarte et al., 2013; Giraud et al., 2008; Jiang et al., 2014; IPCC, 2014).

Changes in carbonate system concentrations in the coastal zone can be conservative or non-conservative (Jiang et al., 2014). Conservative changes arise from the mixing of water masses and from evaporation. The salinity of a water mass is a conservative property and can be used to estimate the conservative component of changes in carbonate system concentrations. The conservative mixing of coastal water masses is often conceptualized as a

10 two-end-member problem; with changes in carbonate system concentrations linearly related to salinity between a freshwater end-member (e.g. rivers, land run-off) and an offshore oceanic end-member (Jiang et al., 2014; Robbins, 2001). Non-conservative changes in the coastal zone are driven by metabolic processes, sediment exchanges and land inputs (Duarte et al., 2013; Jiang et al., 2014). As such, these non-conservative changes can be measured as anomalies in the carbonate system relative to conservative mixing.

The lack of significant freshwater inputs, via rivers and rainfall, in the arid Red Sea means that offshore waters are the only source of mixing exchange to the coastal zone, allowing for the implementation of a single-end-member mixing model (Sofianos and Johns, 2003). A constant oceanic salinity for the offshore region cannot be used to model conservative behaviour, due to basin-scale evaporation which causes a south-north increase in

salinity along the central axis of the Red Sea. The observed south-north increase in alkalinity is smaller than conservative behaviour predicts, due to chemeogenic and biogenic calcium carbonate ($CaCO_3$) precipitation throughout the Red Sea (Jiang et al., 2014; Steiner et al., 2014; Steiner et al., 2018; Wurgaft et al., 2016). Thus, the linear relationships in offshore carbonate system concentrations, combined with the additional variability of coastal evaporation, defines the expected conservative behaviour for the entire coastal zone of the Red Sea (Figure

1).

Here we explore the carbonate system in the eastern (Saudi-EEZ) coastal zone of the Red Sea. We examine concentrations of total alkalinity (TA) and dissolved inorganic carbon (DIC) over and around coral reefs, seagrass meadows and mangrove forests, and compare these to the same properties measured in offshore Red Sea surface

waters. By using a simple single-end-member mixing model, that accounts for conservative changes in the carbonate system of the coastal zone, we detect large ecosystem-driven anomalies in coastal habitats. Smaller non-conservative changes, particularly characteristic of coral reefs, were not able to be detected with high confidence using the over-simplified circulation model but could be resolved with more knowledge of offshore circulation and variability of the carbonate system in the Red Sea.

## 2. Methods

### 2.1 KAUST observations

Between February 2016 and August 2017, seawater samples were collected in the Red Sea during daylight hours from six oceanographic cruises (January/February 2016, January/February 2017, March 2017, April 2017, May 2017, July/August 2017) and at coastal time series stations (Figure 2, Table S1). The six oceanographic cruises visited three types of shallow coastal habitats, spanning the full length of the Saudi-EEZ coast. Open-water samples were also collected on cruises > 50 m upstream from shallow coastal habitats. The coastal time series sampling of surface waters was conducted every two weeks at four stations near the King Abdullah University of Science and Technology (KAUST): a transition water station (22.3093 °N 38.9974 °E, n = 31), a coral reef station (22.25285 °N 38.96122 °E, n = 32), a mangrove forest station (22.3394 °N 39.0885 °E, n = 23) and a seagrass meadow station (22.3898 °N 39.1355 °E, n = 32) (Figure S1).

Transition and offshore water samples were collected using a Niskin bottle deployed off the side of the vessel, together with temperature (T) and salinity (S) recorded with an Ocean Seven 305Plus multi-parameter conductivity-temperature-depth (CTD) instrument. Seawater samples from coastal habitats were collected at a depth > 30 cm below the surface and in close proximity to the habitat. They were collected with a 10 cm diameter by 30 cm long polyvinyl-chloride cylinder, to avoid disturbing the benthic organisms and the associated re-suspension of sediments or epiphytes. The cylinder was carefully moved over the ecosystem and sealed with rubber caps. Measurements of S and T were made at the sampling point using a hand-lowered Ocean Seven 305Plus multi-parameter CTD instrument. The cylinders were then transported to the vessel where water was carefully siphoned using a silicone tube.

Water samples were collected into 12 ml glass vials (DIC) and 50 ml plastic falcon tubes (TA), for all cruises except one (Cruise ID = CSM16) during which TA samples were collected in 12 ml glass vials. To halt biological activity, DIC and TA samples were poisoned to a concentration 0.02 % mercuric chloride solution. TA and DIC were measured at KAUST according to the standard operating procedures as set out by Dickson et al. (2007). DIC was measured by an infrared technique with an Appolo SciTech AS-C3 DIC analyser, and TA was measured by open-cell titration with 0.1 M hydrochloric acid using a Mettler Toledo T50 Autotitrator equipped with an InMotion Pro Autosampler using non-linear curve fitting to determine an equivalence point. Both measurements were standardized using certified reference materials (CRM) obtained from Dr. Andrew Dickson (Scripps Institute of Oceanography). Observations were flagged based on the standard error between replicates, and those that had only single replicates or high standard error (SE > 20 μmol kg$^{-1}$) were excluded from further analysis (n = 17).

### 2.2 WHOI observations

Two oceanographic cruises led by the Woods Hole Oceanographic Institution (WHOI) were conducted in March 2010 and September/October 2011. Targeting open waters of the Red Sea, the cruises traversed the length of the Saudi-EEZ coast. T and S observations were acquired using the ship's CTD, and water samples were collected using Niskin bottles on the CTD rosette. On deck, water samples were transferred into 250 ml glass bottles using

a length of silicone tubing, taking care to minimize exchange with the atmosphere, and were poisoned with 50 µL of a saturated mercuric chloride solution immediately after acquisition. The samples were analysed at WHOI for TA and DIC using a Marianda VINDTA-3C analysis system. TA was determined by non-linear curve fitting of data obtained by open-cell titrations, and DIC concentrations were determined by coulometric analysis, according to the standard operating procedures as set out by Dickson et al. (2007). Both measurements were standardized using CRM obtained from Dr. A. Dickson. The difference between replicate samples averaged 0.6 and 1.5 µmol $kg^{-1}$ for alkalinity and 3.0 and 2.7 µmol $kg^{-1}$ for DIC, for the 2010 and 2011 cruises, respectively.

### 2.3 Published data-sets

Open-water surface observations (< 50 m) collected over 2007-2010 were sourced from published data-sets (Table S1). Data was constrained to a comparable area of the Red Sea in which new observations collected by KAUST and WHOI were obtained (17-28 °N, 30-44 °E). A recent study by Steiner et al. (2018) detected differences between new Red Sea TA observations obtained in 2015, 2016 and 2018, and old Red Sea TA observations from a 1998 cruise on the RV Sea Surveyor (Steiner et al., 2014). We reassess these differences between old and new Red Sea data comparing the new data from Steiner et al. (2018) and from this study with old data collected aboard the RV Sea Surveyor (Steiner et al., 2014) and the RV Marion Dufresne (Papaud and Poisson, 1986). Consequently, we elected to only use data collected within a decade of coastal observations (2007-2017) for our study, as long-term changes were observed in carbonate parameters (Figure 3).

### 2.4 Definition of the coastal zone

The stations used to define the offshore end-member were those (from KAUST, WHOI and published sources) with bottom depths > 200 m according to the General Bathymetry Chart of the Oceans (GEBCO) gridded bathymetry with a 30 arc-seconds resolution (GEBCO_2014 Grid, version 20150318, http://www.gebco.net). All other open-water observations (i.e., stations with bottom depths < 200 m and not collected over a coastal habitat) were labelled as coastal, transition waters. Samples collected over a coastal habitat were classified by the corresponding habitat, either coral reef, seagrass meadow or mangrove forest.

### 2.5 Implementing a single-end-member mixing model

A single-end-member mixing model was used to model conservative TA (cTA) and conservative DIC (cDIC) for coastal observations. First, the distance of a point along the central axis of the Red Sea in km (D) was calculated for each observation. This was done using the "alongTrackDistance" function (default settings) in the R package "geosphere" (Hijmans, 2017) with the reference point 12.7737 °N 43.2618 °E to represent D = 0 and the reference point 28.2827 °N 34.0694 °E to define position of the central south-north axis. The single-end-member model was then implemented by 1) describing the linear variations of offshore S, TA and DIC with D, so that predictions of offshore S ($S_O$), offshore TA ($TA_O$) and offshore DIC ($DIC_O$) can be made from the value of D corresponding to coastal observations, and then 2) calculating cTA and cDIC for observations according to Equations 1-2, which predict the simple dilution and concentration (SDC) effects of evaporation (Figure 1).

Equation 1: $cTA = (S / S_O) * TA_O$

Equation 2: $cDIC = (S / S_O) * DIC_O$

where S is the observed salinity at a coastal observation point and $S_O$, $TA_O$ and $DIC_O$ are calculated from the linear relationships found in step 1, for a distance D corresponding to the observation point.

Other carbonate parameters, the partial pressure of $CO_2$ ($pCO_2$), pH, the saturation state of aragonite ($\Omega_{Ar}$), were calculated with the "carb" function from the R package "seacarb" (Gattuso et al., 2018) which estimates the seawater carbonate system in the absence of borate and sulfate. We employed this function assuming silicate and phosphate concentrations of zero, using $K_1$ and $K_2$ constants from Millero et al. (2010), and using the total scale for pH. Both conservative mixing values and observed values were calculated for other carbonate parameters, from cTA and cDIC, and observed TA and DIC, respectively.

Residual TA (rTA) and residual DIC (rDIC) were then calculated by subtracting cTA and cDIC from observed TA and observed DIC, respectively. Residual of the other carbonate parameters ($rpCO_2$, rpH, $r\Omega_{Ar}$) were calculated by subtracting conservative values of other carbonate parameters (calculated from cTA and cDIC) from observed values of other carbonate parameters (calculated from TA and DIC observations).

## 2.6 Model assumptions and limitations

The single-end-member mixing model assumes simple two-dimensional circulation in a region that exhibits more complex flow. The modelled flow follows a south-north trajectory along the central axis of the Red Sea, with perpendicular coastal flushing from offshore waters located at similar distances along the central axis (Figure 1). This allows changes in the carbonate chemistry of offshore waters, due to both conservative and non-conservative processes, and conservative coastal evaporation to be modelled.

This is a substantial simplification – in fact, the Red Sea has a complex surface flow displaying multiple dynamic eddies along its length (Sofianos and Johns, 2003; Zhan et al., 2014). Depending on the direction of flow, these eddies promote coastal flushing from offshore waters originating further north or further south along the central axis of the Red Sea, mixing in a way the simple single-end-member mixing model cannot capture. Other limitations of the simple single-end-member mixing model include its inability to account for coastal upwelling along the continental shelf, variable mixing of Gulf of Aden waters with Red Sea offshore waters and changes in basin-scale evaporation and calcification which have been documented in previous studies (Anderson and Dyrssen, 1994; Churchill et al., 2014; Krumgalz et al., 1990; Papaud and Poisson, 1986; Steiner et al., 2018).

These limitations cannot be addressed within the present study and would require a sustained observational effort to address knowledge gaps in the carbonate chemistry of the Red Sea, combined with more complex circulation models. Complex circulation models could capture some large-scale variance in circulation, but they are costly simulations, that may still produce questionable results due to the unresolved coastal bathymetry of the Red Sea. Instead, we use the 99 % prediction interval (P.I.) of offshore carbonate chemistry residuals as a bound of model error, and to capture deviations from modelled carbonate chemistry due to the pre-described variations in mixing.

**2.7 Statistical tests**

All statistical tests were performed using R software (R core team, 2017) with a 95 % confidence level. Least squares regression analysis was used on a spatial data subset (all observations excluding time series) to calculate linear relationships with D for S, T and carbonate parameters, thus determining how S, T and carbonate parameters vary along the central axis from south-north. Least-squares regression analysis was also used to calculate relationships between rTA and rDIC. The square of Pearson's correlation coefficient ($r^2$) was used to evaluate the strength of the linear relationships. Least-squares analysis of variance (LS-ANOVA) was used to investigate interaction effects between D and habitat groups to test for significant differences between linear regression slopes with D across habitats for S, T and carbonate parameters.

Seagrass meadows and mangrove forests displayed greater variances compared to other groups (maximum variance / minimum variance > 2) between carbonate parameters, violating the assumption of homoscedasticity between groups required for parametric analysis of variance. For this reason, a Wilcox's robust ANOVA (WR-ANOVA) was chosen to account for heteroscedasticity across habitat groups. WR-ANOVA's for differences in medians were conducted between observations from offshore waters, transition waters and coastal habitats. Tests between medians were chosen, rather than between means, as mangrove habitats displayed skewed TA and DIC observations. Wilcox's robust statistical methods were implemented using the R package "WRS2" (Mair and Wilcox 2018), with the functions "med1way" for testing differences in medians and a bootstrapped t-test employed (Supplementary R Code) for post-hoc analysis.

To assess the strength of seasonal cycles at time series stations and to test differences between habitats, a seasonal proxy (SP) was constructed from T observations at the transition and coral reef time series stations. A cubic smoothing spline, with a smoothing parameter of 0.55, was fit to three iterations of the T seasonal cycle at the stations (Forsythe et al., 1997). The fit was then scaled such that a value of 1 indicates peak summer period, and a value of -1 indicates peak winter period. Parametric tests were chosen to detect correlations with season, as variances across season were roughly homoscedastic. LS-ANOVA was used to assess the significance of seasonal cycle as a predictor in time series observations, to infer the presence of interaction effects between habitats and season in time series observations, and to infer differences in rTA:rDIC slopes between time series observations and spatial observations. WR-ANOVA was also performed on time series observations to assess median differences between the four time series stations.

## 3. Results

### 3.1 The Red Sea offshore end-member

The offshore carbonate system of the Red Sea was characterized along the south-north central axis. Offshore waters exhibited significant and strong (high $r^2$) linear increases in S, TA and DIC along the central south-north axis of the Red Sea as indicated by respective regression analysis with D (Figure 4, Table S3). TA and DIC were normalized to a S of 35 (nTA and nDIC), and both exhibited significant and weak (low $r^2$) linear decreases along the central south-north axis of the Red Sea (Figure 5). However, winter nDIC values appear to deviate from this linear relationship. The nTA and nDIC co-varied along this axis in an average ratio of 0.69 (SE = 0.06, $r^2$ = 0.60, F = 147.7, p < 0.001) nTA to 1 nDIC (Figure 5c). A significant and weak (low $r^2$) linear decrease was found for T against D, that displayed clear seasonal dependencies between summer and winter/spring T (Figure 4). A significant and weak (low $r^2$) increase in pH, a significant and weak (low $r^2$) decrease in $p$CO$_2$, and no significant linear relationship in $\Omega_A$, against D were also observed.

In defining the offshore end-member for implementation in the single-end-member mixing model, offshore observations not representative of the expected linear relationships in the surface offshore Red Sea were removed. These were identified as eleven outlying offshore observations exhibiting a Cook's distance greater than five times the mean in at least one of the three linear models of D, against S, TA and DIC (Figure 1; Cook and Weisberg, 1997). Linear models were then re-fit with the remaining offshore observations (n = 104) to yield Equations 3-5, to be substituted into Equations 1-2 to complete the single-end-member mixing model (Figure 1).

Equation 3: $S_O = 0.00157 * D + 37.47$
Equation 4: $TA_O = 0.0510 * D + 2407$
Equation 5: $DIC_O = 0.0437 * D + 2029$

To approximate the error of the single-end-member mixing model, 99 % prediction intervals (99 % P.I. = mean $\pm$ 2.576 * SD) were calculated by applying the single-end-member mixing model to offshore observations to yield rTA, rDIC, r$p$CO$_2$, rpH and r$\Omega_{Ar}$ (Table S2). These 99 % P.I. represent a cumulative error due to the natural variations of $S_O$, $TA_O$ and $DIC_O$, along with the error propagation associated with the calculations of other carbonate parameters. Two offshore observations used in defining the offshore end-member fell outside the 99 % P.I., both exhibiting high TA, and one exhibiting high DIC.

### 3.2 The Red Sea coastal zone

Coastal observations also displayed significant linear relationships with S from south-north Red Sea (Figure 4). At the transition and coral reef sites, increases in S with D were significant and of comparable strength (indicated by $r^2$ values) to those observed offshore, while a weaker (lower $r^2$) increase in S with D was observed at seagrass meadows (Table S3). No significant increase in S with D was observed at mangrove forests. No interaction effects between habitat type and D were observed for S. This means that rates of increases in S with D did not differ significantly between habitats or offshore waters (excluding mangrove forests; F = 1.55, p = 0.203). Along with

the results of Figure 6a, it can consequently be concluded that, irrespective to D, compared to offshore waters significantly higher median S were observed at mangrove forests across the Red Sea coastal zone. Similarly, it can be concluded that irrespective to D median S for transition waters, coral reefs and seagrass meadows were comparable to the median S observed offshore.

Within coral reefs, seagrass meadows and mangrove forests, decreases in T with D were significant and stronger (higher $r^2$), compared to the decreases observed in offshore waters (Figure 4, Table S4). Tests for interaction effects indicated that rates of change of T with D differed significantly among habitats (F = 7.54, p < 0.001), but these differences were small and did not deviate largely away from those observed in offshore waters (Figure 4).

10 Transition waters displayed similar T to offshore waters along the entire length of the Red Sea. Differently, the three coastal habitats displayed on average slightly higher average T in the southern Red Sea, compared to offshore waters (Figure 4). There was a high sampling bias towards winter/spring in coastal observations and corresponding measurements of in-situ T were not successfully made for all summer observations. Consequently, the seasonal dependencies cannot be confidently compared or described here.

Compared to offshore waters, TA across transition, coral reef and seagrass meadow sites displayed similar rates of increases with D but differing median values and distribution (Figure 4, Figure 6a). Increases in TA with D for each coastal habitat were weaker (lower $r^2$) compared to those observed in offshore waters, with significant linear relationships present for all habitats but mangrove forests. There were no interaction effects between D and habitat

groups (excluding mangrove forests; F = 0.95, p = 0.417). This means that TA at coastal sites displayed similar rates of increase with D compared to offshore waters. Consequently, when also considering results presented in Figure 6a, it can be concluded that transition waters, coral reefs and mangrove forests displayed comparable median TA to offshore waters. Additionally, compared to offshore waters, lower median TA was observed at seagrass meadows across the Red Sea coastal zone (Figure 4, Figure 6a). TA sampled in transition waters and all

coastal habitats exhibited a higher variability compared to that measured offshore.

Increases in DIC with D were also weaker (lower $r^2$) for coastal observations compared to offshore waters, with significant linear relationships observed only for transition and coral reef waters. An interaction effect with D across habitat was observed (excluding mangrove forest and seagrass meadow; F = 4.66, p = 0.011). The

30 significance of this interaction effect was driven by coral reef waters on average displaying lower DIC in the southern Red Sea compared to offshore waters, and comparable DIC in the northern Red Sea (Figure S4a). The variability of coral reef waters was much higher compared to offshore and transition waters. Compared to offshore waters, transition waters showed small increases in median DIC, seagrass meadows showed comparable median DIC and higher variability, whilst mangrove forests displayed higher median DIC with higher variability around

35 this median (Figure 6a).

Observations of pH and $pCO_2$ showed statistically significant, but relatively weak (low $r^2$), linear relationships with D for only seagrass meadow and coral reef sites (Figure 4, Figure 6a). No interaction effects were observed between these habitats and offshore waters. This means that rates of change with D were statistically similar

(excluding transition water and mangrove forest; F = 1.54, 1.98, p = 0.217, 0.141 for pH and $pCO_2$ respectively).

Compared with offshore waters, pH and $pCO_2$ at coral reefs showed statistically similar medians and greater variability. Compared to offshore waters, mangrove forest and seagrass habitats displayed lower median pH and higher median $pCO_2$, and greater variability. Compared to offshore waters, transition observations displayed slightly lower median pH and higher median $pCO_2$, and greater variability. As seen in offshore waters no significant linear relationships with D were observed for $\Omega_{Ar}$ at coastal habitats or transition waters, however coastal observations of $\Omega_{Ar}$ displayed lower medians and higher variability, compared to offshore waters. Mangrove forests displayed the most variability in observed values across all carbonate parameters.

One outlying mangrove forest observation taken near KAUST in 2016 showed high TA values, and low DIC, leading to unrealistic estimations of other carbonate parameters (Figure S2). Further, an isolated mangrove stand was sampled from an inland lake that was tidally flushed (Figure S3a). The two observations taken from this mangrove stand contained much higher TA and DIC compared to observations from coastally residing mangrove stands. The outlying observation was excluded from analysis, however the observations from the inland mangrove stands were not.

### 3.3 Coastal ecosystem anomalies

Using a simple single-end-member mixing model, large non-conservative carbonate system residuals were detected in the coastal Red Sea (Figure 6b). Slopes from least-squares linear regressions with D indicate that non-conservative carbonate system residuals display no significant linear relationships along the south-north central axis of the Red Sea (Table S3, Figure S5). Compared to the 99 % P.I., coral reefs exhibited lower median rDIC and lower median rTA, whilst transition waters exhibited similar median rTA and higher median rDIC, although differences were smallest at these sites. Changes at seagrass meadows and mangrove forests were more pronounced, with a relatively larger variability. Compared to the 99 % P.I., both habitats displayed lower median rTA, however lower median rDIC was observed at seagrass meadows whilst higher median rDIC was observed at mangrove forests.

Non-conservative carbonate system residuals that fall outside of 99 % P.I. deviate significantly away from the expected conservative behaviour of the coastal zone and are concluded to be ecosystem-driven anomalies in the carbonate system (Figure 6b). Transition waters displayed the lowest occurrences of ecosystem-driven anomalies, equally distributed towards higher rTA and lower rTA compared to the 99 % P.I., and mostly towards higher rDIC compared to the 99 % P.I.. Coral reefs also displayed a relatively low range of rTA and rDIC ecosystem-driven anomalies, equally distributed to higher and lower values for rTA compared to the 99 % P.I., and mostly towards lower rDIC compared to the 99 % P.I.. Seagrass meadows and mangrove forests displayed markedly higher occurrences of ecosystem-driven anomalies compared to transition waters and coral reefs. Seagrass meadows displayed ecosystem-driven anomalies distributed mostly towards lower rTA compared to the 99 % P.I., and rDIC ecosystem-driven anomalies distributed mostly towards lower rDIC values compared to the 99 % P.I.. Mangrove forests displayed ecosystem-driven anomalies distributed mostly towards lower rTA compared to the 99 % P.I., and ecosystem-driven anomalies distributed mostly towards higher rDIC compared to the 99 % P.I..

Coastal observations of other carbonate parameters displayed lower median rpH, higher median rpCO$_2$ and lower rΩ$_{Ar}$ compared to the 99 % P.I.. Differences in other carbonate parameters compared to the 99 % P.I. were most pronounced and displayed a large variability in mangrove forests. A significant proportion of ecosystem-driven anomalies in other carbonate parameters was detected at all coastal habitat types. In transition waters, mangrove forests and seagrass meadows, these ecosystem-driven anomalies were mostly observed to have lower pH, higher pCO$_2$ and lower Ω$_{Ar}$ compared to the 99 % P.I.. Compared to the 99 % P.I., coral reefs exhibited a relatively equal distribution of both high and low ecosystem-driven anomalies in pH, a small number of high ecosystem-driven anomalies in pCO$_2$ and low ecosystem-driven anomalies in Ω$_{Ar}$.

## 3.4 Coastal time series

Despite their proximity, there were significant differences in T and S between the three coastal time series sites (Figure 7, Table S6). The coral reef and transition stations displayed similar S of comparable variability, exhibiting no variation with season. Observations of S at the seagrass meadow station were higher, and more variable than those observed at coral reef and transition stations. A seasonal dependency in S was indicated by correlation with the seasonal proxy at this station, however, the correlation is weak and the cycle exhibits a small amplitude. The mangrove forest displayed the highest S, exhibiting no relationship with season and higher variability compared to the coral reef and transition stations.

Strong seasonal dependencies in T were observed at all four time series stations. The seasonal cycles exhibited slower rates of decreases in T towards winter and larger rates of increase in T towards summer. The interaction effect between habitat and the seasonal proxy was significant, indicating that seasonal cycles of T changed between habitat (F = 3.99, p = 0.01). Compared to the transition and coral reef stations, the T observed at the seagrass stations was often higher in winter, and lower in summer, whilst T observed at the mangrove forest station was only higher in summer (Figure 7, Figure S4b).

The coral reef and transition stations displayed a similar series of observations of TA, DIC and their respective residuals. The seagrass meadow station was the only station at which strong, statistically significant seasonal cycles, were observed in both TA and DIC. During summer, the TA and DIC were lower at the seagrass meadow station compared to the transition and coral reef stations; whereas similar TA and DIC were seen at all stations during winter. Similarly, during summer rDIC was lower at the seagrass meadow station compared to that observed at the transition and coral reef stations. In winter rTA and rDIC did not completely return to values observed at the transition and coral reef stations. Weak (low r$^2$), statistically significant seasonal cycles were observed at the seagrass station in pH and pCO$_2$, and at the transition station in DIC, although no clear deviations from other stations exist in these carbonate parameters. Compared to the other three stations, the mangrove forest station displayed no correlations with the seasonal proxy for all carbonate parameters and exhibited much larger variations. TA and DIC at the mangrove forest station were similar to TA and DIC at transition and coral reef stations, indicated by differences in medians. However, rTA and rDIC at the mangrove forest station more closely resembled rTA and rDIC observed at the seagrass station than at the other two stations, as indicated by differences

in medians. No large differences in medians were observed across other carbonate parameters and their respective residuals.

**3.5 Relationship between rTA and rDIC**

Slopes, intercepts and appropriate statistics are presented in Table 1 for linear regression analysis of transition waters, coral reefs, seagrass meadows and mangrove forests. The slope of the relationship between rTA and rDIC was similar between the time series observations and the spatial observations in transition waters, coral reefs and mangrove forests ($F$ = 1.05, 1.11 and 0.07 respectively, $p$ = 0.309, 0.295 and 0.794 respectively) but different in seagrass meadows ($F$ = 6.44, $p$ = 0.014), as indicated by the significance of interaction effects between rDIC across observation subset for each habitat.

Transition water and coral reef observations displayed a significant and weak (low $r^2$) linear relationship between rTA and rDIC with an intercept close to zero. Seagrass meadow observations collected from the time series station displayed significant and strong (high $r^2$) linear relationship between rTA and rDIC, but the spatial subset of observations did not. At the seagrass meadow station there was a significant difference in slope of 0.36 and 0.73 between the spatial and time series subsets respectively, with both regressions displaying similar negative rTA intercepts. The two observations from the inland mangrove stand deviated largely from the extrapolated linear relationships calculated using coastal mangrove stands, and as such were excluded from following regression analysis'. Mangrove forest observations displayed a significant and strong (high $r^2$) linear relationship between rTA and rDIC over both subsets of data, with a negative rTA intercept and a slope of 0.62 and 0.60 for spatial and time series observations, respectively.

## 4. Discussion

The relatively simple oceanography of the offshore Red Sea, with only one oceanic water mass influencing a narrow basin, yields simple linear relationships in salinity (S), temperature (T), total alkalinity (TA), dissolved inorganic carbon (DIC), pH and $pCO_2$ along the south-north central axis (Figure 4). The observed increases in TA

along the central axis of the Red Sea were smaller than would be conservatively predicted from the central axis S data (Figure 5). This is consistent with previous studies which found that basin-scale calcification produces non-conservative deficits of TA that accumulate along the south-north central axis (Jiang et al., 2014; Steiner et al., 2014; Steiner et al., 2018; Wurgaft et al., 2016). The observed increases in DIC were also consistent with basin-scale calcification in summer/spring, however winter results showed more variability around this relationship.

These linear relationships in offshore waters are reflected in the water chemistry of the coastal zone and are removed with the use of a single-end-member mixing model (Figure 1; Figure S5). Doing so enables us to study non-conservative perturbations of carbonate system in shallow benthic habitats at a basin-scale.

The single-end-member conservative mixing model estimates the conservative component of TA and DIC that is

inherent from offshore waters. The variability, or error, of this conservative component is estimated from offshore observations by constructing a 99 % P.I. for rTA and rDIC. This error bound reveals the extent to which habitats alter the carbonate system via ecosystem processes, whist also testing the validity of our single-end-member mixing model. The 99 % P.I. captures offshore variability in S, TA and DIC due to the effects of inter-annual differences, eddies and variable circulation patterns, which act along similar spatial scales in both the offshore

and coastal zones.

We expected to observe only evaporation-driven increases of S in the coastal zone, as freshening by land inputs and precipitation is not thought to be significant. Yet, roughly 25 % of the S observations in coral reefs and transition waters were lower than those observed in offshore waters, highlighting the simplifications inherent in

the single-end-member mixing model (Figure 4, Figure 6a). In winter, this observed freshening could be due to winter precipitation which is accounted for in the model. Alternatively, it could be due to effects that are not captured in the model, including seasonal rivers (wadis) caused by flash floods that occur mainly during October-May (de Vries et al., 2013; Robbins, 2001). These flooding events have not been explored in the context of TA and DIC inputs. In summer, the observed freshening may be due to the influx of Gulf of Aden waters. This

circulation pattern causes cross-shelf variations in surface S along the coast, with salinities in coastal waters observed to be up to 2 units lower than corresponding offshore waters (Churchill et al., 2014; Sofianos and Johns, 2003; Wafar et al., 2016). The implications on the carbonate system of this circulation pattern have also not been characterized. This possibly obscures ecosystem-driven perturbations of rTA and rDIC at coral reefs but has only a small effect on the large ecosystem-driven perturbations observed at mangrove forests and seagrass meadows.

By comparing relative changes in rTA and rDIC in each habitat, inferences can be made regarding the balance of ecosystem processes within Red Sea coastal habitats (Figure 8, Albright et al., 2013; Challener et al., 2016; Cyronak et al., 2018; Gattuso et al., 1998; Zeebe and Wolf-Gladrow, 2001). If a habitat conforms closely to a linear relationship, it can be inferred that the balance of ecosystem processes is relatively uniform across sites.

The slope of the linear relationship indicates the balance of ecosystem processes, with a value determined by the

relative proportions of dominant ecosystem processes represented as directional vectors in Figure 8a. Additionally, the intercepts of the linear relationship are inherited from the signals of upstream ecosystems and the amplitude of an observation along this linear relationship is an indication of a combination of metabolic rate and residence time. It also follows that if a habitat doesn't conform closely to a linear relationship, then the balance of ecosystem processes is variable across sites. These inferences cannot be made with other carbonate parameters as, unlike TA and DIC, they are not invariant to changes in T and pressure. Additionally, other carbon variables respond non-linearly to mixing and variations in TA and DIC. Particularly large but linked changes in TA and DIC in the ratio of roughly 1:1 causes other carbonate parameters to change very little (Zeebe and Wolf-Gladrow, 2001). This effect can be observed at the seagrass meadow time series station with the loss of seasonal cycle in other carbonate parameters (Figure 7).

As our measurements are made from an open-system, the impacts of air-sea $CO_2$ gas exchange on DIC and the subsequent balance between rTA:rDIC cannot be completely discounted. Air-sea $CO_2$ gas exchange may alter the rTA:rDIC relationship away from the balance of ecosystem processes towards negative rDIC. Offshore waters are $pCO_2$ source and further increases in $pCO_2$ in seagrass meadows and mangrove forests will lead to further $CO_2$ outgassing (Figure 6b; Papaud and Poisson, 1986). However, this effect is relatively small compared to the contribution of ecosystem processes when large ecosystem anomalies are present in seagrass meadows and mangrove forests - a result of high, localised metabolic rates (Burkholz et al. 2019; Ho et al. 2014; Sea et al. 2019). Linked measurements of $CO_2$ gas exchange with metabolic fluxes are required to resolve the magnitude of this effect for Red Sea coastal habitats.

All mangrove forests in the Red Sea are comprised of a single species, *Avicennia marina* (Almahasheer et al., 2016) and display a relatively uniform balance of ecosystem processes across the Red Sea (Figure 8e). Both the time series data and the spatial data show statistically similar linear relationships (Figure 8e, Table 1). It follows from this that differences in residence times and metabolic rates are the strongest driver of variability between sites, whilst underlying ecosystem processes remain relatively stable. The positive changes in rTA and rDIC are indicative of high respiration rates, mainly due to high rates of organic matter remineralization, from sediments rich in organic carbon. This lowers pH inducing calcium carbonate dissolution, a mechanism which has been found to be important in previous process-based studies (Burdige & Zimmerman, 2002; Krumins et al., 2013; Meister, 2013; Middelburg et al., 1996). Changes in the negative direction, with deficits in rTA and rDIC, are less expected in mangrove forests, as sediments are rarely net autotrophic (Bouillon et al., 2008; Krumins et al., 2013; Zablocki et al., 2011), but could be inherited from upstream seagrass meadows and coral reefs (Guannel et al., 2016). More support for the latter can be found in time series observations, with TA and DIC at the mangrove forest station conforming closely to those at the seagrass meadow station, whilst also displaying erratic deviations towards high rTA and rDIC that varies similarly to other mangrove forest sites (Figure 7-8). A negative rTA intercept is observed in the mangrove forests linear relationship between rTA and rDIC (Figure 8e), which is consistent with a basin-wide cumulative cross-shelf calcification signal, inherited from upstream coral reefs and potentially even seagrass meadows.

These results suggest that the carbonate system and contributions to air-sea $CO_2$ exchanges of overlying waters in Red Sea mangrove forests is likely significantly mediated by water residence time and mixing, not only by metabolic rates. The inland mangrove forest sampled contained drastically higher TA and DIC in surrounding waters. This resulted in higher $pCO_2$ and a large $CO_2$ source to the atmosphere, compared to coastally-residing mangrove forests (Figure S2-3). De-gassing of $CO_2$, an increase in calcium carbonate dissolution due to decrease in pH, or an increase in redox processes due to oxygen depletion is most-likely the cause for the ratio of rTA:rDIC deviating at this site to an almost perfect 1:1 ratio, from the 0.61 observed elsewhere. This implies that the carbonate system in stagnant water columns over mangrove forests could be different to what is observed in those with variable water exchanges, and the flushing of mangrove forests from surrounding waters can vastly reduce their contribution to air-sea $CO_2$ exchanges. As such, the control of surrounding water exchanges and water residence times should be considered further in these ecosystems, as studies often quantify the influence of mangrove forests on air-sea $CO_2$ exchange using stagnant water columns (Bouillon et al., 2007; Bouillon et al., 2008; Macklin et al., 2019; Sea et al., 2018).

Red Sea seagrass meadows have a high species diversity (Qurban et al., 2019; Kenworthy et al., 2007) and show large ecosystem-driven anomalies in rTA and rDIC, but vary in the balance of ecosystem processes between sites (Figure 8). Within one site, the time series station, a slope of 0.73 is observed, whilst across other sites no strong relationship is found. The variability between rTA and rDIC between sites, and the lack of a significant linear relationship indicates that the balance between ecosystem processes is important in driving the carbonate system of Red Sea seagrass meadows, in combination with metabolic rates and residence times. This is a finding that has been confirmed in separate studies, with the balance of ecosystem processes often effected by variable seagrass meadow density and site oxygenation (Burdige and Zimmerman, 2002; Krumins et al., 2013; Unsworth et al., 2012). Indeed, some seagrass meadow sites visited in the present study have highly variable and species dependent metabolic rates (Anton et al., in review). The slope observed at the seagrass meadow time series site is consistent with the coupling of photosynthesis and sedimentary calcification, promoted by increased pH due to net autrophy in seagrass meadows, which has been shown to result in a ratio of 1:1 change in rTA:rDIC (Barrón et al., 2006; Burdige and Zimmerman, 2002; Krumins et al., 2013; Lyons et al., 2004; Macreadie et al., 2017; Unsworth et al., 2012). Sedimentary sulphur and iron oxidation which, occur alongside sedimentary calcification in oxygenated environments and higher pH, potentially contributes to the lowering of the rTA:rDIC slope below 1 (Burdige and Zimmerman, 2002; Krumins et al., 2013). However, Red Sea seagrass sediments have been observed to contain low levels of iron reducing the posibility of iron oxidation (Saderne et al., 2019; Anton et al., 2018). At the time series station, lower TA and DIC in summer months is due to a combination of increased metabolic rates and/or residence times.

Due to small perturbations in the carbonate system exhibited by transition waters and coral reefs and the uncertainty limits of our model, little can be concluded about the large-scale variability in ecosystem processes in these habitats. Transition waters show few occurrences of ecosystem anomalies inherited from surrounding coastal habitats. Small temporal variations in rTA and rDIC at the coral reef time series station show no linear relationship or seasonal cycle, consistent with variability driven by exchanges through the complex reef system rather than inherent ecosystem processes and metabolic rate. Spatial variability shows similar characteristics, which can be

attributed to a combination of spatial changes in ecosystem processes, residence times, metabolic rates and connectivity (Cyronak et al., 2018; Gattuso et al., 1999; Kleypas et al., 2011; Takeshita et al., 2018). What can be concluded is that coral reef and transition waters have little consequence to air-sea carbon fluxes on a local scale, offering little change in the carbonate system compared to offshore conditions (Figure 4; Figure 6).

The results reported here can offer further explanations to the decadal changes in calcification rates in the Red Sea reported by Steiner et al. (2018), which are also supported within this study (Figure 3). Steiner et al. (2018), reported a $26 \pm 16\%$ decline in total calcium carbonate deposition rate along the basin between 1998 and 2018, concentrated in the southern Red Sea, suggesting that coral reefs in the southern Red Sea are under stress. Warming of the Red Sea has been faster than the global average (Chaidez et al., 2017). This warming has been reported to reduce coral growth rates (Cantin et al., 2010), induce massive bleaching of Red Sea corals south of $20^{\circ}N$ in the summer of 2015 (Hughes et al., 2018, Osman et al., 2018), and lead to the replacement of coral by algal turf. Indeed, this may have reduced carbonate deposition rates in the southern Red Sea over a basin scale, however, our analysis suggests additional contributions to the decline of calcium carbonate deposition in the Red Sea. In particular, mangrove habitats are characterized here as important sites of TA input, likely driven by carbonate dissolution. Hence, the 13% increase in mangrove forests in the Red Sea since 1972 (Almahasheer et al., 2016), could also be reflected in increased rates of carbonate dissolution basin-wide.

## 5. Conclusion

We observed strong evidence of ecosystem-driven perturbations in the carbonate system over Red Sea coastal habitats. We employed a simple single-end-member mixing model to estimate the expected conservative behaviour over the coastal zone of the Red Sea. We find that 1) along-shelf changes in TA and DIC exhibit strong linear relationships that are consistent with net basin-scale calcium carbonate precipitation; 2) ecosystem-driven changes in TA and DIC are larger than offshore variations in >70% of sampled seagrass meadows and mangrove forests, changes which are influenced by a combination of longer water residence times and community metabolic rates; and 3) the sampled mangrove forests show strong and consistent contributions from both organic respiration and other sedimentary processes (carbonate dissolution and secondary redox processes), while seagrass meadows display more variability in the relative contributions of photosynthesis and other sedimentary processes (carbonate precipitation and oxidative processes). With the available data we cannot conclude if differences in magnitude of rTA and rDIC within habitats reflect differences in residence times or metabolic rates. The results of this study highlight the importance of resolving the influences of water residence times, mixing and upstream habitats on mediating the carbonate system and coastal air-sea $CO_2$ fluxes over coastal habitats in the Red Sea.

## Code Availability

Code for the calculation of the Wilcox Robust ANOVA post-hoc test can be found in the supplementary material. All other code relating to figures and analysis was constructed in R (version 3.4.3) and is available upon request to the corresponding author.

## Data Availability

The full data set used in this study can be obtained from PANGAEA (doi: [10.1594/PANGAEA.899850](10.1594/PANGAEA.899850)).

## Author Contribution

K.B. collected a portion of the KAUST samples, performed most of the sample analysis on KAUST samples, performed data analysis, developed conceptual ideas and wrote the manuscript. V.S. collected a portion of the KAUST samples, performed some sample analysis on KAUST samples and contributed to manuscript preparation. D.C.M. and J.H.C. collected and analysed WHOI samples and contributed to concept development and manuscript preparation. S.A. facilitated the collection of samples at three time series stations. C.M.D contributed to the concept development and to manuscript preparation.

## Competing Interests

The authors of this manuscript declare no competing interests.

## Acknowledgments

This research was funded by King Abdullah University of Science and Technology (KAUST) through baseline funding and competitive center funding to Carlos M. Duarte, and Susana Agusti (BAS/1/1072-01-01). The sample collection and laboratory analysis of KAUST observations was carried out by K. Baldry and V. Saderne. The sample collection and laboratory analysis of Woods Hole Oceanographic Institution (WHOI) observations was supported by Award Nos. USA-00002, KSA-00011 and KSA-00011/02 made by KAUST to WHOI. We thank all of the contributors who were responsible for collecting and managing Red Sea data that was openly sourced. The full data set used in this study can be obtained from PANGAEA (doi: [10.1594/PANGAEA.899850](10.1594/PANGAEA.899850)). Supplementary tables and figures can be found in the supporting information accompanying this manuscript. We thank A. K. Gusti, A. Anton, M.L. Berumen, P. Carrillo-de-Albornoz, D.J. Cocker, N. Garcias-Bonet, C.K. Martin, J. Martinez-Ayala and S. Overmans for their help and support in sample collection and CTD deployments towards this study. Finally, we thank Z. Steiner and one anonymous reviewer for their useful comments that contributed to the development of the manuscript.

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

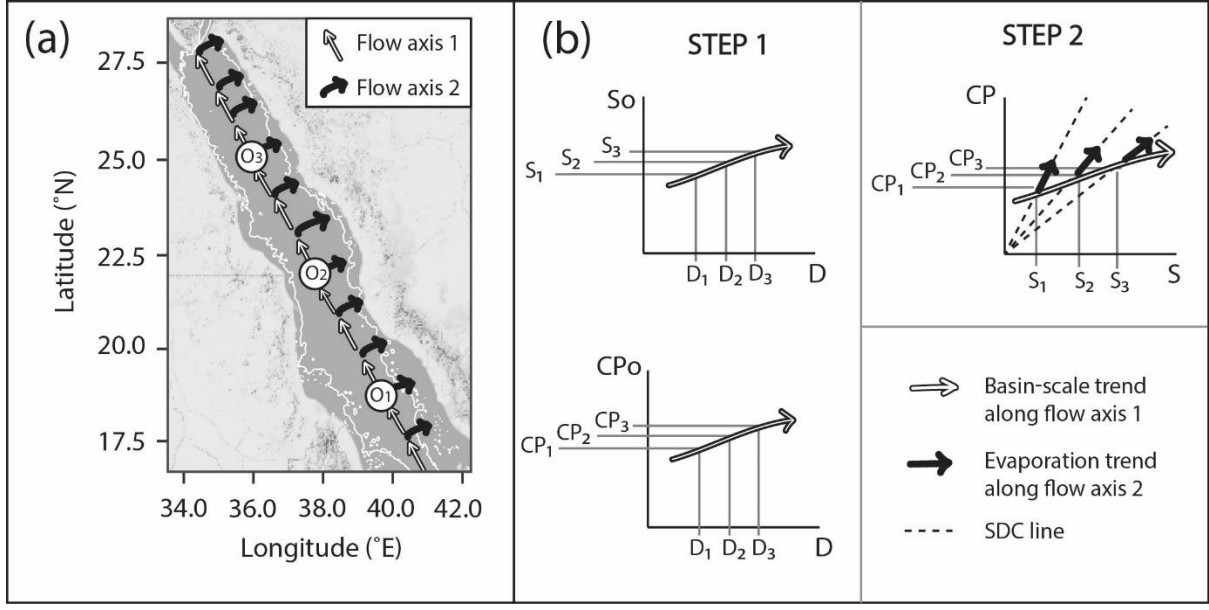

**Figure 1.** Schematic of the single-end-member mixing model used in the present study. Panel (a) displays the assumed circulation pattern which has two flow axes overlaid on a map of the Red Sea produced using © Stamen Design LLC and R software. Flow axis 1 is along the south-north central axis where waters experience cumulative changes due to basin-scale evaporation and calcification. Flow axis 2 is perpendicular to this axis, where it is assumed that evaporative effects prevail as waters transition from offshore locations ($O_i$) to coastal regions. The thin white line indicates the 200m bathymetry. Panel (b) explains the single-end-member mixing model in two steps, to determine conservative estimates of a carbonate parameter (CP: TA or DIC) for the coastal zone. $O_i$ represents an offshore end-member at a location in offshore waters lying along flow axis 1 (the central axis) at distance $D_i$, from a fixed reference point in the southern Red Sea, and corresponding salinity $S_i$ and carbonate parameter measurement $CP_i$, as derived from basin-scale linear relationships. $CP_i$ is then scaled along the simple dilution and concentration (SDC) line to obtain coastal estimates for carbonate parameters.

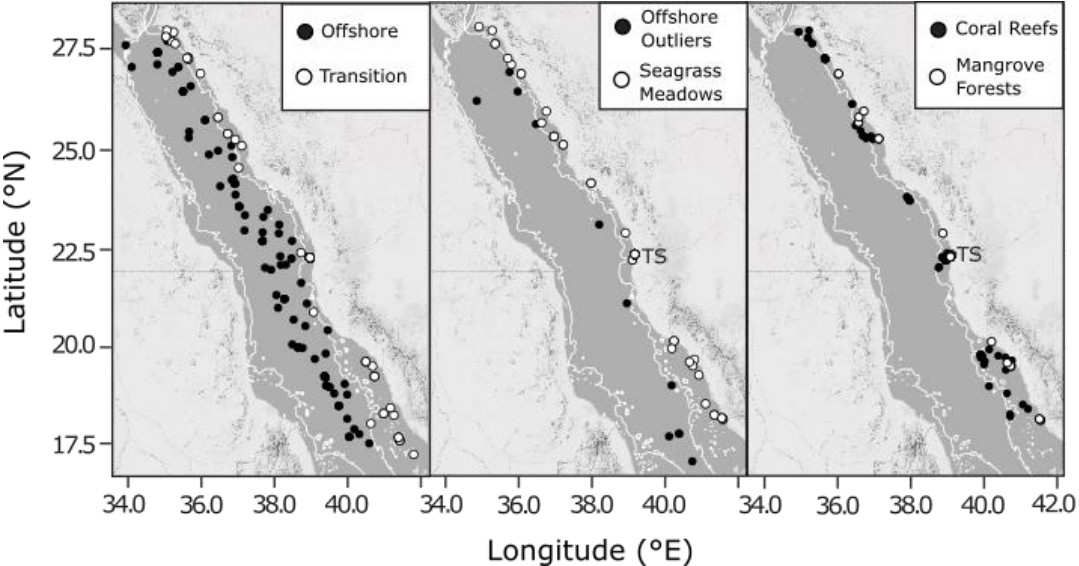

**Figure 2.** The spatial distribution of combined observations from the Red Sea data sets used in the present study, shown against a 200m bathymetry boundary (thin white line) overlaid on maps of the Red Sea produced using © Stamen Design LLC and R Software. Observations are classified as offshore, transition, seagrass meadows, coral reefs or mangrove forests. Time series stations are indicated as TS (Figure S1). Outliers identified in offshore observations are also shown.

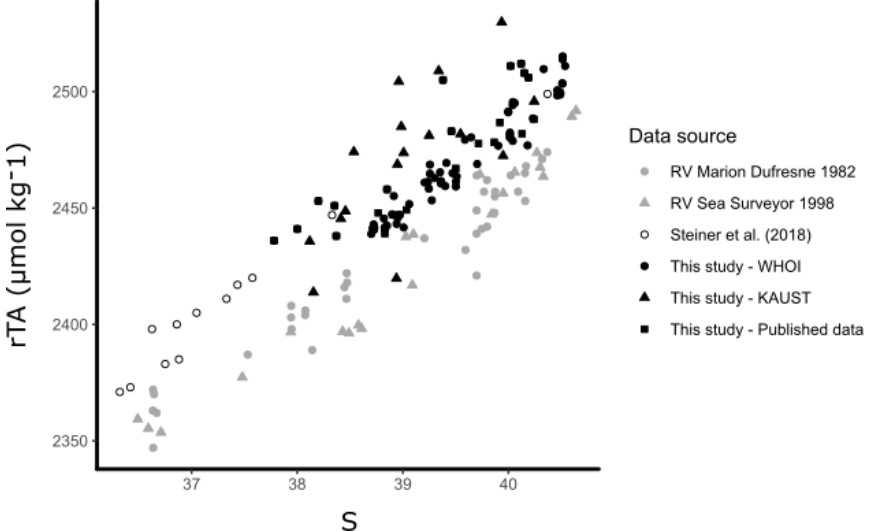

**Figure 3.** Offshore observations of TA are shown against S, as in Steiner et al. (2018), to illustrate the difference between old (grey) and new (black) observations of carbonate chemistry in the offshore surface waters of the Red Sea. The observations presented from this study are only from offshore waters (excluding outliers).

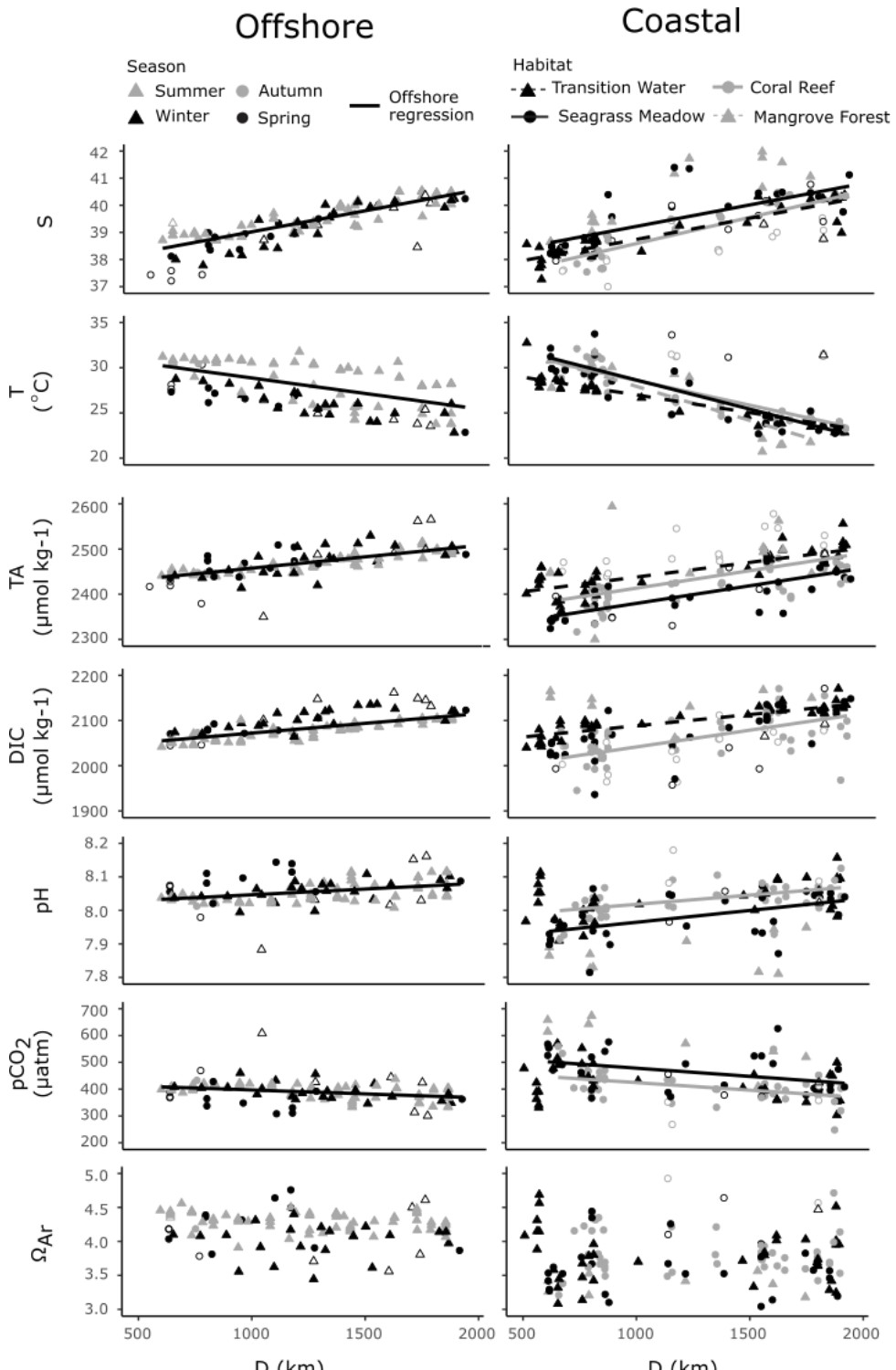

**Figure 4.** Offshore observations of S, T and carbonate parameters (left) and four coastal habitats (right) are presented against distance along the south-north central axis (D), on the same scales. Significant linear regressions for all combinations of variables are drawn as lines, with associated statistics reported in Table S3. Offshore outliers were not included in determining offshore regressions against D. Coastal observations from time series stations and an outlying mangrove observation were not included in regressions against D. Note that not all coastal

observations are displayed, an expanded scale is shown in Figure 6. Hollow symbols indicate offshore outliers (right panel) and coastal summer observations (left panel).

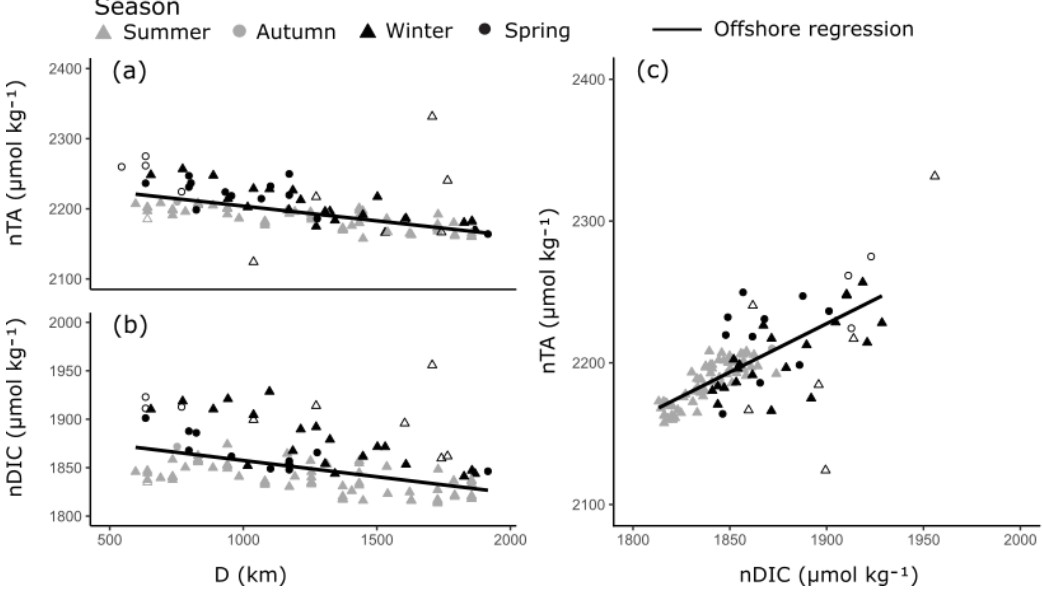

**Figure 5.** Linear relationships in nTA and nDIC along the south-north central axis of the Red Sea (a-b), and between nTA and nDIC (c) in the offshore waters. Symbols indicate the season in which samples were collected. All linear relationships were statistically significant and offshore outliers were not included in the linear regressions (hollow symbols).

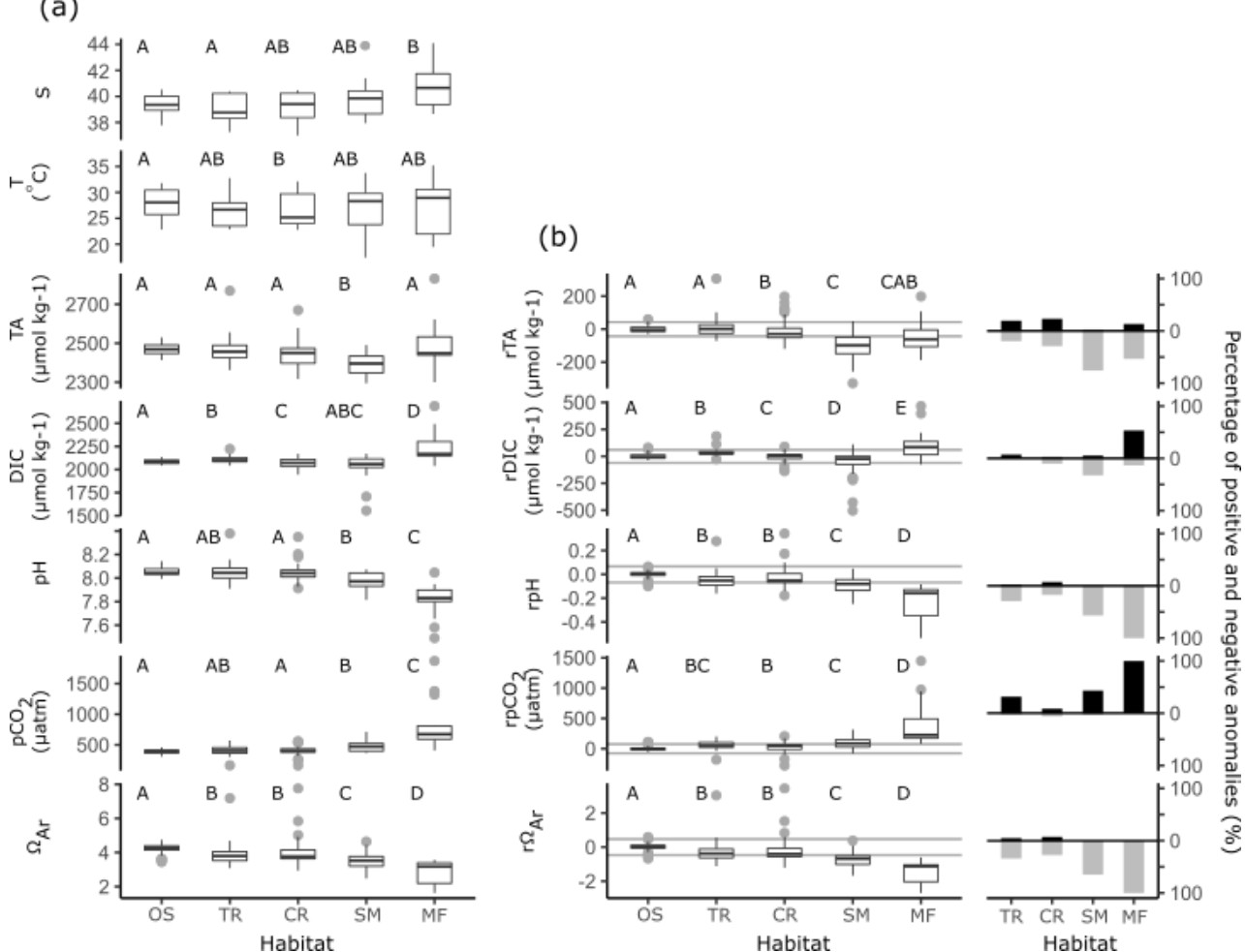

**Figure 6.** Box-plot distributions of (a) observed S, T, carbonate parameters and (b) non-conservative carbonate system residuals which are presented by habitat group: offshore (OS), transition waters (TR), coral reef (CR), seagrass meadow (SM) and mangrove forest (MF). Each box-plot displays the median, the first and third quartiles, and whiskers that extend to the mean +/- 1.5 times the inter-quartile range. Grey dots represent observations that extend outside the whiskers of the boxplot. Grey lines in panel (b) indicate the upper and lower bounds of the 99% P.I. defined by offshore observations. The proportion of ecosystem anomalies (%) observed in both the positive and negative directions are presented alongside, and to the right of boxplots in panel (b) (Table S4). Grouping letters (A-D) assigned above boxplots indicate the results of post-hoc bootstrapped t-tests, summarized from statistics presented in Table S5. If tests showed significant similarities at the 0.05 significance level with another habitat across a variable they were assigned the same letter.

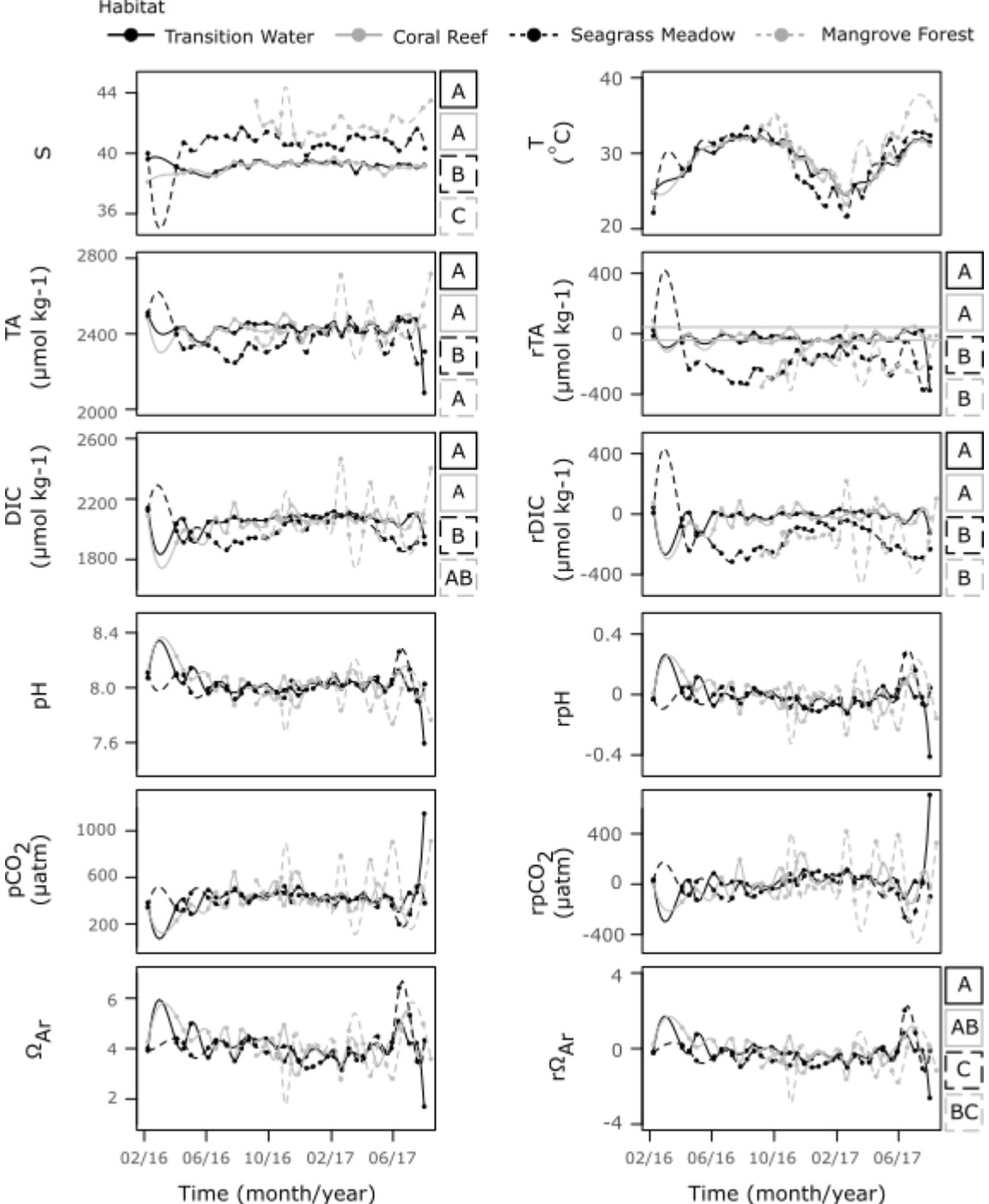

**Figure 7.** Time series observations of S, T, carbonate parameters and non-conservative carbonate system residuals collected from the four time series stations. Observations are shown fitted with a spline function of order 100 by the method of Forsythe et al., (1977). For variables which displayed a significant result for WR-ANOVA tests for differences in medians across habitat groups, results from post-hoc bootstrapped t-tests are shown as letters (A-C) to the right of the plot. If tests showed significant similarities at the 0.05 significance level with another habitat across a variable they were assigned the same letter.

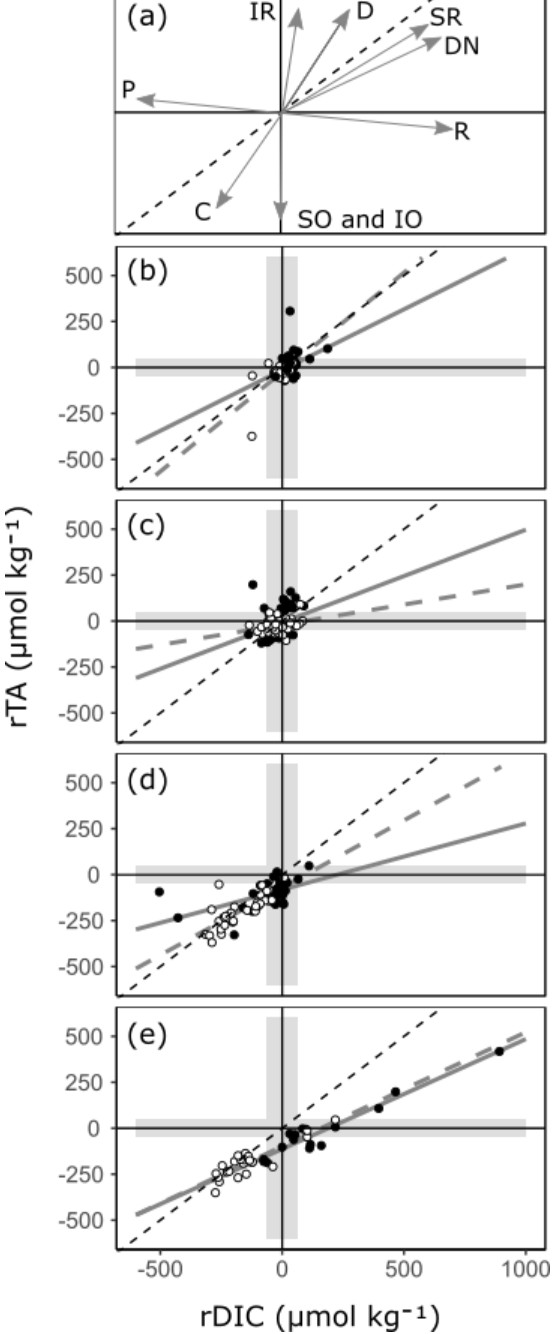

**Figure 8.** A reference plot (a) showing unitless directional vectors of change in the rTA vs. rDIC space for multiple ecosystem processes. Below, observations of rTA vs. rDIC from transition water and the three coastal habitats is presented: (b) transition water, (c) coral reef, (d) seagrass meadow and (e) mangrove forest. Time series observations are indicated with open circles, and all other spatial data is indicated with closed circles. A 1:1 reference line is shown in all plots (black dashed line) as well as regression lines ($r^2 > 0.6$) for the time series subset (grey dashed line) and the spatial subset (grey solid line). The reference plot includes directional vectors for calcium carbonate precipitation (C), calcium carbonate dissolution (D), primary production (P), respiration (R), iron reduction (IR), sulphate reduction (SR), denitrification (DN), sulphur oxidation (SO) and iron oxidation (IO). The shaded envelope represents the calculates 99% P.I. for rTA and rDIC. An expanded figure of panel (e) showing inland mangrove stands is presented in Figure S2b.

5    **Tables**

**Table 1:** Intercept (±SE), slope (±SE), correlation coefficient ($r^2$), F statistics (F) and p-values (p) of linear regressions of rTA versus rDIC for different subsets of coastal observations.

| Data subset | Intercept | Slope | $r^2$ | F | p |
|---|---|---|---|---|---|
| Transition Water | -24.6 (±7.5) | 0.91 (±0.17) | 0.29 | 27.8 | <0.001 |
| Coral Reef | -11.3 (±6.1) | 0.39 (±0.13) | 0.08 | 8.55 | 0.004 |
| Seagrass Meadow: Times series | -71.4 (±20.8) | 0.73 (±0.11) | 0.60 | 45.26 | <0.001 |
| Seagrass Meadow: Spatial | -81.9 (±12.4) | 0.36 (±0.09) | 0.32 | 15.8 | <0.001 |
| Mangrove Forest: Time series | -98.7 (±13.2) | 0.62 (±0.07) | 0.79 | 77.26 | <0.001 |
| Mangrove Forest: Spatial | -113.2. (±13.4) | 0.60 (±0.05) (0.86 with inland stands) | 0.92 | 161.8 | <0.001 |