# Peer review of "Anomalies in the Carbonate System of Red Sea Coastal Habitats"

_Biogeosciences, 2019_

## Referee Comment (RC1) · Zvi Steiner (Referee) · 4 Jul 2019

Zvi Steiner (Referee)

zs313@cam.ac.uk

Baldry et al., report analyses of alkalinity and dissolved inorganic carbon along the main axis of the Red Sea and into some of the region's coastal ecosystem. These measurements are used to assess the magnitude of changes in total alkalinity and dissolved inorganic carbon in the various ecosystems of the Red Sea. The Red Sea has an exceptionally long stretch of tropical coastal habitats that are under increasing pressure globally. The unique oceanographic conditions of this region, e.g. relatively simple flow regime, high salinity and high temperatures turn the Red Sea into a very relevant site for studying how changes in different environmental variables affect coral reefs, mangroves and seagrass meadows. It is also a region that was historically very poorly represented in oceanographic studies.

[Figure]

This paper provides an important dataset which is an essential addition to the data currently available in the scientific literature. I think that the discussion of this data could be made substantially stronger if it will be better tied to previous publications and used to explain changes that were observed in the carbonate system of the Red Sea. As noted by the authors, there has be a large increase in the total alkalinity of the Red Sea surface waters in recent years (Steiner et al., 2018). Previous publications on this topic were limited in their ability to assess if this change was only due to changes in coral calcification and ecology or there has been a shift in other ecosystems as well, and whether or not these correlate with each other. The authors chose to ignore half of their dataset and focus exclusively on the older samples but I think that comparisons between old and new trends of ecosystem specific rTA and rDIC could be valuable. Together with comparison with past data regarding the change in DIC and total alkalinity of the central Red Sea axis, this can potentially provide a test for the various hypotheses previously suggested for the cause of the reduction in Red Sea calcification rates.

A few specific comments:

Please refrain from using the shortcuts OCP and D. They are not intuitive and had me going back to check their meaning several times.

Fig. 3: please indicate in the figure legend if these are surface waters only.

Fig. 6: I don't understand from the legend what A, B, C, AB etc. stand for. It needs to be explained in the paper, not in the appendix.

Fig. 7: From which year is the data presented here?
* * *

---

## Referee Comment (RC2) · Anonymous Referee #2 · 4 Jul 2019

General comments: In this paper, Baldry et al. combine carbon measurements from the open ocean and east coastal areas in the Red Sea to model ecosystem-driven changes on the carbon system of coral reefs, mangrove forest, and seagrass meadows. In this region, oceanographic studies in general as well as carbon and ecosystem studies are heavily underrepresented, despite its extreme conditions regarding hydrography and vulnerable ecosystems. The paper by Baldry et al. represent an important contribution to the biogeochemical research from the Red Sea, and by using novel data, historical data, and a model tool, they increase our knowledge about driving forces for coastal ecosystems.

More specific comments: The word "trend" is used but the word refers to change over time, and you do not use the word this way. As I understand, you simply mean linear

relationship between e.g. offshore salinity and distance from a point in the southern Red Sea, or alkalinity and the mentioned distance.

Every now and then you put up statements and explain them later in the manuscript, e.g. P2, L22 about linear trends, P4, L18 where you introduce D without explaining it until later in the text. I encourage you to gather the statement and explanation, to make the reading easier.

You refer to numerous interesting papers, please include a separation between ";" and the following author name. This comment is valid for the whole paper. E.g. L 28: (Bauer et al., 2013; Camp et al., 2016; Cyronak et al., 2018; Gattuso et al., 1998; Guannel et al., 2016; Unsworth et al., 2012) – here I have added space.

You discuss several limitations with the single-end-member model, but you actually did choose this model. Please add an argument stating why, despite all its limitations, you made this decision.

You use the words strong or weak linear increase when you actually mean high or low r2. Just be aware that strong/weak linear increase might also be understood as a line with high or low slope.

Detailed comments: P1 L16: you introduce the word Ấntrend", which refer to change over time. But this is not what you mean, right? Rather use "linear relationship"

P2 L11: I suggest a more direct language: As such, these non-conservative changes can be measured as anomalies from the carbonate system which has experienced conservative mixing. L22: You state "The linear trend in offshore carbonate system concentrations…" without explaining or showing what you mean by this. Again, I suspect that you mean simply linear relationship and not trend. L23: suggest to not use the word "norm" but only "expected conservative behaviour"

P3 L27: second last word: switch "a" with "an" L27: please add if this method also use non-linear curve fitting L29: add full address of Dr. A. Dickson the first time he

is mentioned L35: which type of CTD is used L36: I guess you used a plastic tube to transfer the water from the water samples to the glass bottle?

P4 L1: add reference for VINDTA-3C L11: references for long-term changes: it seems like you have older refs than Steiner et al. 2018, please add L17: add the word "observed" so the sentence reads "describing the linear variations of observed S, TA and DIC …" L19: define D here (distance from a defined zero point in the southern Red Sea) L19: explain difference between observed S, TA and DIC and predictions of So, Tao, and DICo (both along the north south axis). Why don't you use observed offshore values in Eq 1 and 2? L31: "All other open waters" means 200m< transition<coastal? And would you please define coastal?

P5 L2: what is an "observed estimate" L6: to ensure clarity, add "coastal" to "observed TA and observed DIC" L18: change "two-end-member" with "single-end-member" L20: as above L29: suggest changing "linear trends with D for S …variables" with "how S, temperature and carbonate variables vary along the central axis from south to north". L30: add "to" between the words "used investigate"

P7 L3: suggest a simpler language: "The offshore carbonate system of the Red Sea was characterized along the south-north …". L4: suggest "Offshore waters exhibited significant and strong (high r2) linear increase in S …". This sequence of words should be use all over, because the words "strong" or "weak" are connected to the linearity and only indirectly to significance. L6, L9, L10, L11: as above Are the strong/weak linear trend values summarized in a Table? If so, this should be announced early in paragraph 3.1. L32: you describe the "Coastal observations", but then the word "central axis" should be exchanged with something else, since the coast is not along the central axis. Maybe just use "from south to north". L37: change "end-member" with "waters"

P8 L7, L10, L14, L18, L19, L22 and more: as above. In general, I advise you to not use "end-member" when you mean "saters". It is just confusing. L31: do you really mean "trend", if so, over which time. If not, change with "linear relationship"

P9 L15: you are comparing to a "norm", are you referring to anomalies (P2, L12) or expected conservative behaviour (P2, L23)? L18: as above L22: delete "norm or" L24 and 25 ant the rest of this and next paragraphs: use other words for "norm"

P10 L15: you write "seasonal trend", do you actual mean "seasonal variation"? If so, change all over P21 L5: change "The latitude at which time series stations are at is indicated by the text"Ts"" with "Time series stations are indicated as TS". Refer to Figure S1 in the figure text

P22, Figure 2: define Oi earlier in figure text. L7: change "estimate" with "determine" L9: after "central axis at distance Di" add "from a fixed reference point in the southern Red Sea"

P23, Figure 3: L3: suggest text " Offshore observations of S, T and carbon variables (left) and four coastal . . ."

P24, Figure 4: L3: change "end-members" with "waters" L4: after "included" add "in the"

P26, Figure 6: A, B, C, D, AB, BC, CD are not explained

Table S2: in the footnote you use a * as a multiplicator, this is confusing since the same sign is used as footnote numbering. Please change Table S3 and S4: please include units where you can (T, TA, DIC, pCO2 etc). Table S4: change title from "By Habitat descriptive statistics for carbon variable habitat groups for all coastal . . ." to "Descriptive statistics for carbon variable habitat groups for all coastal . . ." Table S5: add units where feasible Table S7: as above

---

## Author Comment (AC1) · 31 Aug 2019

Author Response to Reviewer Comment 2: Anonymous Reviewer 2

Key:

- Review comment is in **bold**
- Author response is in normal text
- Changes made are in *italics*

We thank Reviewer 2 for their contribution to the review process of this paper. We have considered the reviewers comments carefully and incorporated their feedback in the below dialogue.

**General comments: In this paper, Baldry et al. combine carbon measurements from the open ocean and east coastal areas in the Red Sea to model ecosystem-driven changes on the carbon system of coral reefs, mangrove forest, and seagrass meadows. In this region, oceanographic studies in general as well as carbon and ecosystem studies are heavily underrepresented, despite its extreme conditions regarding hydrography and vulnerable ecosystems. The paper by Baldry et al. represent an important contribution to the biogeochemical research from the Red Sea, and by using novel data, historical data, and a model tool, they increase our knowledge about driving forces for coastal ecosystems.**

**More specific comments:**

**The word "trend" is used but the word refers to change over time, and you do not use the word this way. As I understand, you simply mean linear relationship between e.g. offshore salinity and distance from a point in the southern Red Sea, or alkalinity and the mentioned distance.**

This is correct. We are describing linear relationships and not trends with time

*The terms "trends" and "linear trends" have been replaced with "linear relationships", except when used in the context of seasonal changes. Here, the term "seasonal trend" has been changed to "seasonal dependency"*

**Every now and then you put up statements and explain them later in the manuscript, e.g. P2, L22 about linear trends, P4, L18 where you introduce D without explaining it until later in the text. I encourage you to gather the statement and explanation, to make the reading easier.**

We agree these issues need be addressed. We have altered the identified problems as outlined in the specific comments, in particular introducing and explaining D. The f*irst paragraph of 2.4(now 2.5) now reads:*

*"A single-end-member mixing model was used to model conservative TA (cTA) and conservative DIC (cDIC) for coastal observations. First, the distance of a point along the central axis of the Red Sea in km (D) was calculated for each observation. This was done using the "alongTrackDistance" function (default settings) in the R package "geosphere" (Hijmans, 2017) with the reference point 12.7737°N 43.2618°E to represent D = 0 and the reference point 28.2827°N 34.0694°E to define position of the central south-north axis. The single-end-member model was then implemented by 1) describing the linear variations of offshore S, TA and DIC with D, so that predictions of offshore S ($S_O$), offshore TA ($TA_O$) and offshore DIC ($DIC_O$) can be made from the value of D corresponding to coastal observations, and then 2) calculating cTA and cDIC for observations according to Equations 1-2, which predict the simple dilution and concentration (SDC) effects of evaporation (Figure 2)."*

**You refer to numerous interesting papers, please include a separation between ";" and the following author name. This comment is valid for the whole paper. E.g. L 28: (Bauer et al., 2013; Camp et al., 2016; Cyronak et al., 2018; Gattuso et al., 1998; Guannel et al., 2016; Unsworth et al., 2012) – here I have added space.**

Noted.

*A space has been added between ";" and the following author name.*

**You discuss several limitations with the single-end-member model, but you actually did choose this model. Please add an argument stating why, despite all its limitations, you made this decision.**

Noted.

*We have adjusted section 2.6 as follows:*

*2.6 Model Assumptions and Limitations*

*The single-end-member mixing model assumes simple two-dimensional circulation in a region that exhibits more complex flow. The modelled flow follows a south-north trajectory along the central axis of the Red Sea, with perpendicular coastal flushing from offshore waters located at similar distances along the central axis (Figure 2). This allows changes in the carbonate chemistry of offshore waters, due to both conservative and non-conservative processes, and conservative coastal evaporation to be modelled.*

*It is well known that this is not the case and the Red Sea has a complex surface flow displaying multiple dynamic eddies along its length (Sofianos and Johns, 2003; Zhan et al., 2014). Depending on the direction of flow, these eddies promote coastal flushing from offshore waters originating further north or further south along the central axis of the Red Sea, mixing in a way the simple single-end-member mixing model cannot capture. Other limitations of the simple single-end-member model include its inability to account for coastal upwelling along the continental shelf, variable mixing of Gulf of Aden waters with Red Sea offshore waters and changes in basin-scale evaporation and calcification which have been documented in previous studies (Anderson and Dyrssen, 1994; Churchill et al., 2014; Krumgalz et al., 1990; Papaud and Poisson, 1986; Steiner et al., 2018).*

*These limitations cannot be addressed within the present study and require a sustained observational effort to address knowledge gaps in the carbon chemistry of the Red Sea, combined with more complex circulation models. Complex circulation models could capture some large-scale variance in circulation, but they are costly simulations that may still produce questionable results due to the unresolved coastal bathymetry of the Red Sea. Instead, we use the 99% P.I. of offshore carbonate chemistry residuals as a bound of model error, and to capture deviations from modelled carbonate chemistry due to variations in circulation.*

**You use the words strong or weak linear increase when you actually mean high or low r2. Just be aware that strong/weak linear increase might also be understood as a line with high or low slope.**

We have attempted to make the association between strong/weak to r2 in the text by referencing to r2 values in parentheses more often. We have also edited section "2.7 Statistical tests" to read

"All statistical tests were performed using R software (R core team, 2017) with a 95% confidence level. Least squares regression analysis was used to calculate linear relationships with D for S, temperature and carbonate variables, thus determining how S, temperature and carbonate variables vary along the central axis from south to north. Least squares regression analysis was also used to

calculate relationships between rTA and rDIC. The square of Pearson's correlation coefficient ($r^2$) of linear relationships was used to evaluate the strength of the relationships."

**Detailed comments:**

**P1 L16: you introduce the word "trend", which refer to change over time. But this is not what you mean, right? Rather use "linear relationship"**

Addressed above

**P2 L11: I suggest a more direct language: As such, these non-conservative changes can be measured as anomalies from the carbonate system which has experienced conservative mixing.**

Thank you.

*This suggestion has been taken throughout the paper, and the use of the term "norm" removed from the text throughout.*

**L22: You state "The linear trend in offshore carbonate system concentrations..." without explaining or showing what you mean by this. Again, I suspect that you mean simply linear relationship and not trend.**

Addressed above.

**L23: suggest to not use the word "norm" but only "expected conservative behaviour"**

Changed.

*"norm" replaced with "expected conservative behaviour"*

**P3 L27: second last word: switch "a" with "an"**

Changed.

*"a" replaced with "an"*

**L27: please add if this method also use non-linear curve fitting**

Changed.

*"and TA was measured by open-cell titration with 0.1 M hydrochloric acid using a Mettler Toledo T50 Autotitrator equipped with an InMotion Pro Autosampler" to "and TA was measured by open-cell titration with 0.1 M hydrochloric acid using a Mettler Toledo T50 Autotitrator equipped with an InMotion Pro Autosampler using non-linear curve fitting to determine an equivalence point"*

**L29: add full address of Dr. A. Dickson the first time he is mentioned**

Addressed

*"Dr. A. Dickson" changed to "Dr. Andrew Dickson (Scripps Institution of Oceanography)" on first occurrence*

**L35: which type of CTD is used**

We cannot address this point; we do not know the make and model of the ship's CTD. We do not see this as a critical lack.**L36: I guess you used a plastic tube to transfer the water from the water samples to the glass bottle?**

Yes. Words added to text.

**P4 L1: add reference for VINDTA-3C**

The text states the make (Marianda) and model (VINDTA-3C) of the instrument that was used. There is no "reference" for the instrument.

**L11: references for long-term changes: it seems like you have older refs than Steiner et al. 2018, please add**

This is the only reference showing long-term changes in the Red Sea. Older references include older Red Sea data or some other aspect of Red Sea carbonate chemistry, but the aim of those studies was not to show long-term changes.

*No changes made*

**L17: add the word "observed" so the sentence reads "describing the linear variations of observed S, TA and DIC ..."**

Changed.

*"describing the linear variations of S, TA and DIC ..." to "describing the linear variations of observed S, TA and DIC ..."*

**L19: define D here (distance from a defined zero point in the southern Red Sea)**

Changed.

*D is now defined earlier.*

*First paragraph of 2.4 now reads:*

*"A single-end-member mixing model was used to model conservative TA (cTA) and conservative DIC (cDIC) for coastal observations. First, the perpendicular distance of a point along the central axis of the Red Sea in km (D) was calculated for each observation. This was done using the "alongTrackDistance" function (default settings) in the R package "geosphere" (Hijmans, 2017) with the reference point 12.7737°N 43.2618°E to represent D = 0 and the reference point 28.2827°N 34.0694°E to define position of the central south-north axis. The single-end-member model was then implemented by 1) describing the linear variations of S, TA and DIC with D, so that predictions of offshore S ($S_O$), offshore TA ($TA_O$) and offshore DIC ($DIC_O$) can be made from D corresponding to coastal observations, and then 2) calculating cTA and cDIC for coastal observations according to Equations 1-2, which predict the simple dilution and concentration (SDC) effects of coastal evaporation (Figure 2)."*

**L19: explain difference between observed S, TA and DIC and predictions of So, Tao, and DICo (both along the north south axis). Why don't you use observed offshore values in Eq 1 and 2?**

It is true we do use offshore observations in the methods to calculate 99% P.I. Thus the specification of only coastal observations being used in the model has been removed.

*Section 2.4 (now 2.5) now reads:*

*2.5 Implementing a single-end-member mixing model*

*A single-end-member mixing model was used to model conservative TA (cTA) and conservative DIC (cDIC) for coastal observations. First, the perpendicular distance of a point along the central axis of*

*the Red Sea in km (D) was calculated for each observation. This was done using the "alongTrackDistance" function (default settings) in the R package "geosphere" (Hijmans, 2017) with the reference point 12.7737°N 43.2618°E to represent D = 0 and the reference point 28.2827°N 34.0694°E to define position of the central south-north axis. The single-end-member model was then implemented by 1) describing the linear variations of observed offshore S, TA and DIC with D, so that predictions of offshore S ($S_O$), offshore TA ($TA_O$) and offshore DIC ($DIC_O$) can be made from D corresponding to coastal observations, and then 2) calculating cTA and cDIC for observations according to Equations 1-2, which predict the simple dilution and concentration (SDC) effects of evaporation (Figure 2).*

*Equation 1: $cTA = (S/S_O)*TA_O$*

*Equation 2: $cDIC = (S/S_O)* DIC_O$*

*Where S is the observed salinity at a coastal observation point and $S_O$, $TA_O$ and $DIC_O$ are calculated for a distance D corresponding to the observation point from the linear relationships found in step 1.*

*Other carbon parameters, the partial pressure of $CO_2$ ($pCO_2$), pH, the saturation state of aragonite ($\Omega_{Ar}$), were calculated with the R package "seacarb" (Gattuso et al. 2018) assuming silicate and phosphate concentrations of zero, employing the total scale for pH and using the carbonate constants from Millero et al. (2010). Both conservative values and observed values were calculated for other carbon parameters, from cTA and cDIC, and observed TA and DIC, respectively.*

*Residual TA (rTA) and residual DIC (rDIC) were then calculated by subtracting cTA and cDIC from observed TA and observed DIC, respectively. Residual other carbon parameters ($rpCO_2$, rpH, $r\Omega_{Ar}$) were calculated by subtracting conservative values of other carbon parameters (calculated from cTA and cDIC) from observed values of other carbon parameters (calculated from TA and DIC observations).*

**L31: "All other open waters" means 200m< transition<coastal? And would you please define coastal?**

*A new section 2.4 has been added into the manuscript:*

*2.4 Definition of the coastal zone*

*Offshore observations used to describe the offshore end-member were those (from KAUST, WHOI and published sources) with bathymetry > 200 m below sea-level according to the General Bathymetry Chart of the Oceans (GEBCO) gridded bathymetry with a 30s resolution (BODC, https://www.bodc.ac.uk/). All other open-water observations not collected over a coastal habitat were labelled as coastal, transition waters. Samples collected over a coastal habitat were classified by the corresponding habitat, either coral reef, seagrass meadow or mangrove forest.*

**P5 L2: what is an "observed estimate"**

Changed to "observed value"

*Now reads:*

*Other carbon parameters, the partial pressure of $CO_2$ ($pCO_2$), pH, the saturation state of aragonite ($\Omega_{Ar}$), were calculated with the R package "seacarb" (Gattuso et al. 2018) assuming silicate and phosphate concentrations of zero, employing the total scale for pH and using the carbonate constants from Millero et al. (2010). Both conservative values and observed values were calculated for other carbon parameters, from cTA and cDIC, and observed TA and DIC, respectively.*

*Residual TA (rTA) and residual DIC (rDIC) were then calculated by subtracting cTA and cDIC from observed TA and observed DIC, respectively. Residual other carbon parameters (rpCO₂, rpH, rΩₐᵣ) were calculated by subtracting conservative values of other carbon parameters (calculated from cTA and cDIC) from observed values of other carbon parameters (calculated from TA and DIC observations).*

**L6: to ensure clarity, add "coastal" to "observed TA and observed DIC"**

Not changed as coastal specification removed as above.

**L18: change "two-end-member" with "single-end-member"**

Thankyou

*Changed "two-end-member" to "single-end-member"*

**L20: as above**

Thank you

*Changed "two-end-member" to "single-end-member"*

**L29: suggest changing "linear trends with D for S ...variables" with "how S, temperature and carbonate variables vary along the central axis from south to north".**

The sentence has been adjusted to make it more explicit what linear relationships with D mean.

*This sentence now reads:*

*Least squares regression analysis was used to calculate linear relationships with D for S, temperature and carbonate variables, thus determining how S, temperature and carbonate variables vary along the central axis from south to north*

**L30: add "to" between the words "used investigate"**

Thank you!

*"to" added between "used" and "investigate"*

**P7 L3: suggest a simpler language: "The offshore carbonate system of the Red Sea was characterized along the south-north ...".**

Thank you!

*P7 L3 now reads:*

*The offshore carbonate system of the Red Sea was characterized along the south-north central axis.*

**L4: suggest "Offshore waters exhibited significant and strong (highr2) linear increase in S ...". This sequence of words should be use all over , because the words "strong" or "weak" are connected to the linearity and only indirectly to significance.**

Noted.

*Changed "....strong (high $r^2$), significant...." to ".... Significant and strong (high $r^2$)....." (and variations with "weak") everywhere.*

**L6, L9, L10, L11: as above Are the strong/weak linear trend values summarized in a Table? If so , this should be announced early in paragraph 3.1.**

Noted.

*Table S3 reference added at the start of Section 3.1*

**L32: you describe the "Coastal observations", but then the word "central axis" should be exchanged with something else, since the coast is not along the central axis. Maybe just use "from south to north".**

Noted.

*Sentence now reads:*

*Coastal observations also displayed significant linear relationships with S from south-north along the Red Sea (Figure 3-5).*

**L37: change "end-member" with "waters"**

Noted.

*Changed "end-member" to "waters"*

**P8L7, L10, L14, L18, L19, L22andmore: as above. In general, I advise you to not use "end-member" when you mean "saters". It is just confusing.**

Noted.

*Changed "end-member" to "waters" in section 3.2 and where appropriate in the text. "offshore end-member" is now only used when referring to the single-end-member mixing model and not when comparing to offshore observations.*

**L31: do you really mean "trend", if so, over which time. If not, change with "linear relationship"**

Addressed above

**P9 L15: you are comparing to a "norm", are you referring to anomalies (P2, L12) or expected conservative behaviour (P2, L23)?**

Comparing to the expected conservative behaviour using a 99% P.I. for error

*Changed to 99% P.I.*

**L18: as above**

As above

**L22: delete "norm or"**

Changed.

**L24 and 25 ant the rest of this and next paragraphs: use other words for "norm"**

Changed.

*Here we are comparing to 99% P.I. so we have now changed the language to directly say this by replacing "norm" with "99% P.I.".*

**P10 L15: you write "seasonal trend", do you actual mean "seasonal variation"? If so, change all over**

Addressed above.

*Changed to "seasonal dependency"*

**P21 L5: change "The latitude at which time series stations are at is indicated by the text"Ts"" with "Time series stations are indicated as TS". Refer to Figure S1 in the figure text**

Changed as suggested.

**P22, Figure 2: define Oi earlier in figure text.**

We have mentioned Oi earlier in the text

*Now reads:*

*"……Flow axis 1 is along the south-north central axis where waters experience cumulative changes due to basin-scale evaporation and calcification. Flow axis 2 is perpendicular to this axis, where it is assumed that evaporative effects prevail as waters transition from offshore locations (Oi) to coastal regions."*

**L7: change "estimate" with "determine"**

Changed as suggested.

**L9: after "central axis at distance Di" add "from a fixed reference point in the southern Red Sea"**

Changed as suggested.

**P23, Figure 3: L3: suggest text " Offshore observations of S, T and carbon variables (left) and four coastal ..."**

Changed as suggested.

**P24, Figure 4: L3: change "end-members" with "waters"**

Changed as suggested.

**L4: after "included" add "in the"**

Changed as suggested.

**P26, Figure 6: A, B, C, D, AB, BC, CD are not explained**

Noted.

*Figure 7 caption changed from "Grouping letters indicate the results of post-hoc bootstrapped t-tests, summarized from statistics presented in Table S5. If tests showed significant similarities at the 0.05 significance level with another habitat across a variable they were assigned the same letter." to "Grouping letters (A-D) assigned above boxplots indicate the results of post-hoc bootstrapped t-tests, summarized from statistics presented in Table S5. If tests showed significant similarities at the 0.05 significance level with another habitat across a variable they were assigned the same letter."*

**Table S2: in the footnote you use a * as a multiplicator, this is confusing since the same sign is used as footnote numbering.**

Noted

*Changed footnotes* [+]

**Please change Table S3 and S4: please include units where you can (T, TA, DIC, pCO2 etc).**

Noted.

*Added unit references. Also added unit references to Table S2.*

**Table S4: change title from "By Habitat descriptive statistics for carbon variable habitat groups for all coastal ..." to "Descriptive statistics for carbon variable habitat groups for all coastal ..."**

Noted.

*Reviewer suggested change implemented.*

**Table S5: add units where feasible**

Noted.

*Added unit references*

**Table S7: as above**

Noted.

*Added unit references*

---

## Author Comment (AC2) · 2 Sep 2019

Author response to Reviewer Comment 1: Zvi Steiner

Key:

- Review comment is in **bold**
- Author response is in normal text
- Changes made are in *italics*

We thank Dr. Zvi Steiner for their contribution to the review process of this paper. We have considered the reviewers comments carefully and incorporated their feedback in the below dialogue.

**Baldry et al., report analyses of alkalinity and dissolved inorganic carbon along the main axis of the Red Sea and into some of the region's coastal ecosystem. These measurements are used to assess the magnitude of changes in total alkalinity and dissolved inorganic carbon in the various ecosystems of the Red Sea. The Red Sea has an exceptionally long stretch of tropical coastal habitats that are under increasing pressure globally. The unique oceanographic conditions of this region, e.g. relatively simple flow regime, high salinity and high temperatures turn the Red Sea into a very relevant site for studying how changes in different environmental variables affect coral reefs, mangroves and seagrass meadows. It is also a region that was historically very poorly represented in oceanographic studies.**

**This paper provides an important dataset which is an essential addition to the data currently available in the scientific literature. I think that the discussion of this data could be made substantially stronger if it will be better tied to previous publications and used to explain changes that were observed in the carbonate system of the Red Sea. As noted by the authors, there has be a large increase in the total alkalinity of the Red Sea surface waters in recent years (Steiner et al., 2018).**

**Previous publications on this topic were limited in their ability to assess if this change was only due to changes in coral calcification and ecology or there has been a shift in other ecosystems as well, and whether or not these correlate with each other. The authors chose to ignore half of their dataset and focus exclusively on the older samples but I think that comparisons between old and new trends of ecosystem specific rTA and rDIC could be valuable.**

**Together with comparison with past data regarding the change in DIC and total alkalinity of the central Red Sea axis, this can potentially provide a test for the various hypotheses previously suggested for the cause of the reduction in Red Sea calcification rates.**

We thank the reviewer for the positive comments and the guidance provided. We focused our study in offshore Red Sea data published in the literature that is less than 10 years old, as well as new offshore data collected within the scope of the study to train our offshore model and calculate ecosystem specific rTA and rDIC. We noted a trend in TA and DIC in our offshore data compared to old cruises, which support the findings of Steiner et al. 2018. We are taking the reviewers comments on board and adding an extra subplot to Figure 4 to confirm the differences between old and new data with an independent dataset from 1982. This data will be made openly available via PANGAE upon acceptance of the manuscript.

[Figure]

We also note that we did not include data reported in Steiner et al. (2018) in our published dataset. We have now added this data to our analysis, increasing the transition water dataset from 71 to 72 and the offshore dataset from 92 to 101. The inclusion has not changed the results or main findings of the paper substantially, and we will be happy to work this into our revised manuscript. A revised Section 3.1 and Table S2 is shown below to illustrate the minor changes the adding in the Steiner et al. (2018) data has to our model.

**3.1 The Red Sea offshore end-member**

*The offshore carbonate system of the Red Sea was characterized along the south-north central axis. Offshore waters exhibited significant and strong (high $r^2$) linear increases in S, TA and DIC along the central south-north axis of the Red Sea as indicated by respective regression analysis with D (Figure 3, Table S3). TA and DIC were normalized to a salinity of 35 (nTA and nDIC), and both exhibited significant and weak (low $r^2$) linear decreases along the central south-north axis of the Red Sea (Figure 4). However, winter nDIC values appear to deviate from this linear relationship. The nTA and nDIC co-varied along this axis in an average ratio of 0.87 (SE= 0.07, $r^2 = 0.60$, F = 147.7, p<0.001) nTA to 1 nDIC (Figure 4c). A significant and weak (low $r^2$) linear decrease was found for T against D, that displayed clear seasonal dependencies between summer and winter/spring temperatures (Figure 3). A significant and weak (low $r^2$) increase in pH, a significant and weak (low $r^2$) decrease in pCO$_2$, and no significant linear relationship in $\Omega_A$, against D were also observed.*

*In defining the offshore end-member for implementation in the single-end-member mixing model, offshore observations not representative of the expected linear relationships in the surface offshore Red Sea were removed. These were identified as eleven outlying offshore observations exhibiting a Cook's distance greater than five times the mean in at least one of the three linear models of D, against S, TA and DIC (Figure 1; Cook and Weisberg, 1997). Linear models were then re-fit with the remaining offshore observations (n = 104) to yield Equations 3-5, to be substituted into Equations 1-2 to complete the single-end-member mixing model (Figure 3).*

*Equation 3: $S_O = 0.00157*D + 37.47$*

*Equation 4: $TA_O = 0.0510*D + 2407$*

*Equation 5: $DIC_O = 0.0437*D + 2029$*

*To approximate the error of the single-end-member mixing model, 99% prediction intervals (99% P.I. = mean ± 2.576*sd) were calculated by applying the single-end-member mixing model to offshore observations to yield rTA, rDIC, rpCO₂, rpH and $r\Omega_{Ar}$ (Table S2). These 99% P.I represent a cumulative error due to the natural variations of $S_O$, $TA_O$ and $DIC_O$, along with the error propagation associated with the calculations of other carbon parameters. Two offshore observations used in defining the offshore end-member fell outside the 99% P.I., both exhibiting high TA, and one exhibiting high DIC.*

**Table S2:** *Defining statistics of the normal error for residual carbon variable estimates, as calculated*

*from offshore observations*

| | Residual mean | Residual standard deviation | Lower 99% P.I. bound[+] | Upper 99% P.I. bound[++] | % offshore observations outside the 99% P.I. (excluding/including outliers) |
|---|---|---|---|---|---|
| rTA (μmol/kg) | 0 | 16.79 | -43.25 | 43.25 | 1.1/5.9 |
| rDIC (μmol/kg) | 0 | 23.33 | -60.09 | 60.09 | 2.2/5.9 |
| rpH | $-5 \times 10^{-4}$ | $2.69 \times 10^{-2}$ | $-6.97 \times 10^{-2}$ | $6.87 \times 10^{-2}$ | 4.3/6.9 |
| rpCO₂ (μatm) | 0.10 | 30.20 | -77.69 | 77.90 | 4.3/7.9 |
| $r\Omega_{Ar}$ | 0.0006 | 0.1879 | -0.4833 | 0.4845 | 4.3/6.9 |

Whereas, the long-term trends observed in the Red Sea are not the focus of this study, we do believe that adding an element in the discussion using our new results to provide some insights onto the long-term changes noted by Steiner et al. (2018), will add value to the paper, as suggested by the reviewer. We have, therefore, added the following paragraph in the discussion section:

*"The results reported here can offer explanation to the decadal changes in calcification rates in the Red Sea reported by Steiner et al. (2018), which are also supported by inspection of the data compiled*

*here (Figure 4a). Steiner et al. (2018), reported a 26 ± 16% decline in total CaCO₃ deposition rate along the basin between 1998 and 2018, concentrated in the southern Red Sea, suggesting that coral reefs in the southern Red Sea are under stress. Indeed, warming of the Red Sea, which has been faster than the global average (Chaidez et al. 2017), has been reported to reduce coral growth rates (Cantin et al. 2010), and massive bleaching of Red Sea corals south of 20°N in the summer of 2015 (Hughes et al. 2018, Osman et al. 2018), and replacement by algal turf, may have reduced carbonate deposition rates in the southern Red Sea further. Our analysis suggests additional contributions to decline carbonate deposition in the Red Sea. In particular, mangrove habitats are characterized here as important sites of carbonate dissolution. Hence, the 13% increase in mangrove forests in the Red Sea over the past 30 years (Almahasheer et al. 2016), is expected to have resulted in increased rates of carbonate dissolution basin-wide."*

Commented [CMD1]:

Commented [CMD2]:

Commented [CMD3]:

Commented [CMD4]:

Additional references.

Cantin, N. E., Cohen, A. L., Karnauskas, K. B., Tarrant, A. M. & McCorkle, D. C. Ocean warming slows coral growth in the central Red Sea. Science 329, 322–325 (2010).

Osman, E.O., Smith, D.J., Ziegler, M., Kürten, B., Conrad, C., El-Haddad, K.M., Voolstra, C.R. and Suggett, D.J., 2018. Thermal refugia against coral bleaching throughout the northern Red Sea. Global change biology, 24(2), pp.e474-e484.

Hughes, T. P. et al. Spatial and temporal patterns of mass bleaching of corals in the Anthropocene. Science 359, 80–83 (2018).

Chaidez, V., Dreano, D., Agusti, S., Duarte, C.M. and Hoteit, I., 2017. Decadal trends in Red Sea maximum surface temperature. Scientific reports, 7(1), p.8144.

**A few specific comments:**

**Please refrain from using the shortcuts OCP and D. They are not intuitive and had me going back to check their meaning several times.**

Noted. We will remove the acronym OCP. However, we need to retain the acronym D, as it is a key model parameter that we define in the methods and in Figure 2. We have edited the methodology to make what D more obvious to the reader by defining it first in the single end-member model.

*OCP replaced with other carbon parameter. First paragraph of 2.4 now reads:*

*"A single-end-member mixing model was used to model conservative TA (cTA) and conservative DIC (cDIC) for coastal observations. First, the perpendicular distance of a point along the central axis of the Red Sea in km (D) was calculated for each observation. This was done using the "alongTrackDistance" function (default settings) in the R package "geosphere" (Hijmans, 2017) with the reference point 12.7737°N 43.2618°E to represent D = 0 and the reference point 28.2827°N 34.0694°E to define position of the central south-north axis. The single-end-member model was then implemented by 1) describing the linear variations of S, TA and DIC with D, so that predictions of offshore S (S_O), offshore TA (TA_O) and offshore DIC (DIC_O) can be made from D corresponding to coastal observations, and then 2) calculating cTA and cDIC for coastal observations according to Equations 1-2, which predict the simple dilution and concentration (SDC) effects of coastal evaporation (Figure 2)."*

*Caption edited from "…….. O_i represents a location in the offshore end-member lying along the central axis at distance D_i, ….." to "…….. O_i represents a location in the offshore end-member lying along flow axis 1 (the central axis) at distance D_i, ….."*

**Fig. 3: please indicate in the figure legend if these are surface waters only.**

Noted.

*"Observations of S, T and carbon variables in the offshore end-member (left)" changed to*
*"Observations of S, T and carbon variables in the surface offshore end-member (left)"*

**Fig. 6: I don't understand from the legend what A, B, C, AB etc. stand for. It needs to be explained in the paper, not in the appendix.**

Noted.

*Figure 7 caption changed from "Grouping letters indicate the results of post-hoc bootstrapped t-tests, summarized from statistics presented in Table S5. If tests showed significant similarities at the 0.05 significance level with another habitat across a variable they were assigned the same letter." to "Grouping letters (A-D) assigned above boxplots indicate the results of post-hoc bootstrapped t-tests, summarized from statistics presented in Table S5. If tests showed significant similarities at the 0.05 significance level with another habitat across a variable they were assigned the same letter."*

**Fig. 7: From which year is the data presented here?**

2016/2017.

*Figure 7 x-label changed to month/year (below). Mm/dd is a mistake. Thank you!*

[Figure]

---

## Referee Report (RR1)

Referee comments to the manuscript
**Anomalies in the Carbonate System of Red Sea Coastal Habitats by Kimberlee Baldry, Vincent Saderne, Daniel C. McCorkle, James H. Churchill, Susana Agusti and Carlos M. Duarte**

I have read the revised manuscript with interest and my comments now are mostly connected to wordings. My first comment is about being consistent.
Throughout the manuscript, you use both the terms "carbon parameters" and "carbon variables", please be consistent. I suggest "carbon variables".
Throughout the manuscript, you use both the terms "carbonate chemistry" and "carbon chemistry", please be consistent.
Throughout the manuscript, you mix "salinity" and "S", please be consistent. The same also for "T" and "temperature".
Regarding depths, please be consistent using space or not when you write a depth/distance in the text, e.g. 50m or 50 m.
Your use two abbreviations for Offshore ("O" and "OF"), two for Transition waters ("T" and "TR"), and the same for Coral reefs, Seagrass meadows and Mangrove forest. Please be consistent.
You use both "time series" and "time-series", please be consistent.

My second comment is on readability. The content is fine, but the text is very heavy to read due to the length of sentences. I advise you to use split most of the long sentences to make the life easier for the reader.

Detailed comments:
P3 L7: is it possible to indicate a distance (m or km?).
P4 L12: change "2015/6" to "2015-2016".
P4 L13: change "Steiner et al. 2014» to «Steiner et al., 2014».
P4 L14: add «data from» just before «the RV Sea Surveyor».
P4 L22: check unit "s".
P5 L2: use "w" in the first word and not "W".
P5 L8: replace "carbon constants" with "$K_1$" and $K_2$", and add information about HSO4 constant and borate constant.
P5 L12: replace "Residual other" with "Residuals of the other".
P7 L16: please remove "(n=11)", since this is mentioned in the following sentence.
P8 L26: please explain the "interaction effect".
P8 L32: replace "increased" with "increases"
P9 L9: replace "late" with "lake"
P10: there are four time series stations, three coastal and one in the transition zone, right? The title of this chapter is "Coastal Time Series", so why do you also mention the transition time series? An option could be to change the title to "Time Series".
P10 L31: add "to the" after "compared".
P12 L2: replace "end-member" with "water mass".
P12 L9: replace "varience" with "variability".
P12 L15: replace "end-member" with "water".
P12 L14-16: please re-write.
P12 L16: delete one of the "in"'s.
P12 L20: replace "observations of S" with "the salinity observations".
P13 L2-3: the statement that DIC is only affected by mixing and metabolic processes is only true if your data is sub-surface data, but if your data is from the surface or close to surface, you also have to consider air-sea gas exchange. This brings me to the fact that there is no information about which depths the samples are collected from. Such information is vital, please include

P21 L10: delete "at is".

P22 L7-8: delete "and the transition from offshore to coastal waters".

P25 L4: include the word "which" between "residuals" and "are".

P26 L4: replace "Forsythe, Malcolm and Moler" with "Forsythe et al."

P27 L3: remove the first "and".

Table S1: For Steiner15 data, please write out the years: 2015-2016, and not 2015/6

Table S3: explain numbers in parenthesis

Table S3: intercept S of coral reef 3.667, should be 36.67?

Table S5: Please rewrite the table text

---

## Author Response (AR2)

**Authors Response**

Anomalies in the Carbonate System of Red Sea Coastal Habitats

Kimberlee Baldry, Vincent Saderne, Daniel C. McCorkle, James H. Churchill, Susana Agusti and Carlos M. Duarte

We thank the editors for inviting us to resubmit our article, and the reviewers who took the time to read and comment on it. Within this document we have included:

**List of relevant changes**

- Paragraph on air-sea $CO_2$ flux in discussion
- Added more symbols to Figure 3
- Updated and added another interaction plot Figure S3

**Point-by -point response to Zvi Steiner**

**Author response to Z. Steiner**

The authors have responded in **bold.**

I recommend publication of the revised manuscript by Baldry et al., subject to few minor changes:

Page 15, 2nd line: Almahasheer et al. compared their newer satellite images to images from 1972, this is more than 30 years ago.

**Have changed to: Hence, the 13% increase in mangrove forests in the Red Sea since 1972 (Almahasheer et al., 2016), could also be reflected in increased rates of carbonate dissolution basin-wide.**

Fig. 3:
Is the data plotted from this study from WHOI, KAUST or both? The different datasets were obtained in different years and by different analytical methods so I think you should use different symbols to present them.

**We have added more symbols to the plot, to delineate different samples from our study.**

I assume that all samples presented in Fig. 3 are surface water samples? Please state that in the caption.

**Caption changed to:**

**"Figure 3. Offshore observations of TA are shown against S, as in Steiner et al. (2018), to illustrate the difference between old (grey) and new (black) observations of carbonate chemistry in the offshore surface waters of the Red Sea. The observations presented from this study are only from offshore waters (excluding outliers)."**

Fig. 3 is first cited in the text before Fig. 2., please change the order of figures.

**Changed.**

**Point-by -point response to Anonymous  Reviewer**

**Author response to anonymous reviewer**

The authors have responded in **bold**

Referee comments to the manuscript Anomalies in the Carbonate System of Red Sea Coastal Habitats by Kimberlee Baldry, Vincent Saderne, Daniel C. McCorkle, James H. Churchill, Susana Agusti and Carlos M. Duarte

I have read the revised manuscript with interest and my comments now are mostly connected to wordings.

My first comment is about being consistent. Throughout the manuscript, you use both the terms "carbon parameters" and "carbon variables", please be consistent. I suggest "carbon variables". Throughout the manuscript, you use both the terms "carbonate chemistry" and "carbon chemistry", please be consistent. Throughout the manuscript, you mix "salinity" and "S", please be consistent. The same also for "T" and "temperature". Regarding depths, please be consistent using space or not when you write a depth/distance in the text, e.g. 50m or 50 m. Your use two abbreviations for Offshore ("O" and OF"), two for Transition waters ("T" and "TR"), and the same for Coral reefs, Seagrass meadows and Mangrove forest. Please be consistent. You use both "time series" and "time-series", please be consistent.

**Thankyou the much-appreciated attention to detail. We have corrected the text based on your comments and criticised it for more inconsistencies.**

My second comment is on readability. The content is fine, but the text is very heavy to read due to the length of sentences. I advise you to use split most of the long sentences to make the life easier for the reader.

**Thankyou, we have attempted to use shorter sentences, as suggested, to increase readability.**

Detailed comments: P3 L7: is it possible to indicate a distance (m or km?).

**Altered to "Open-water samples were also collected on cruises > 50 m upstream from shallow coastal habitats."**

P4 L12: change "2015/6" to "2015-2016".

**Changed to 2015, 2016 and 2018**

P4 L13: change "Steiner et al. 2014» to «Steiner et al., 2014».

**Changed, and corrected other similar formatting errors.**

P4 L14: add «data from» just before «the RV Sea Surveyor».

**We have amended this sentence. It now reads:**

**"We reassess these differences between old and new Red Sea data comparing the new data from Steiner et al. (2018) and from this study with old data collected aboard the RV Sea Surveyor (Steiner et al., 2014) and the RV Marion Dufresne (Papaud and Poisson, 1986)."**

P4 L22: check unit "s".

**Changed to "arc-seconds"**

P5 L2: use "w" in the first word and not "W".

**Changed.**

P5 L8: replace "carbon constants" with "K1" and K2", and add information about HSO4 constant and borate constant.

**The paragraph has been edited and replaced with**

"**Other carbon parameters, the partial pressure of $CO_2$ ($pCO_2$), pH, the saturation state of aragonite ($\Omega_{Ar}$), were calculated with the "carb" function from the R package "seacarb" (Gattuso et al., 2018) which calculates the seawater carbonate system in the absence of borate and sulfate. We employed this function assuming silicate and phosphate concentrations of zero, using $K_1$ and $K_2$ constants from Millero et al. (2010), and using the total scale for pH. Both conservative mixing values and observed values were calculated for other carbon parameters, from cTA and cDIC, and observed TA and DIC, respectively.**"

P5 L12: replace "Residual other" with "Residuals of the other".

**Changed.**

P7 L16: please remove "(n=11)", since this is mentioned in the following sentence.

**Changed.**

P8 L26: please explain the "interaction effect".

**Paragraph now reads:**

"**Increases in DIC with D were also weaker (lower $r^2$) for coastal observations compared to offshore waters, with significant linear relationships observed only for transition and coral reef waters. An interaction effect with D across habitat was observed (excluding mangrove forest and seagrass meadow; F = 4.66, p = 0.011). The significance of this interaction effect was driven by coral reef waters on average displaying lower DIC in the southern Red Sea compared to offshore waters, and comparable DIC in the northern Red Sea (Figure S4a). The variability of coral reef waters was much higher compared to offshore and transition waters. Compared to offshore waters, transition waters showed small increases in median DIC, seagrass meadows showed comparable median DIC and higher variability, whilst mangrove forests displayed higher median DIC with higher variability around this median (Figure 6a).**"

P8 L32: replace "increased" with "increases"

**Changed.**

P9 L9: replace "late" with "lake"

**Changed.**

P10: there are four time series stations, three coastal and one in the transition zone, right? The title of this chapter is "Coastal Time Series", so why do you also mention the transition time series? An option could be to change the title to "Time Series".

**We have elected to keep "Coastal Time Series" as the title of this chapter, as the transition time series site is coastal. Throughout the paper we have used the term coastal to describe waters sampled on the shelf of the Red Sea, or surrounding offshore islands – those with a bottom water**

**depth < 200 m. The transition waters are coastal waters, that have not been sampled in close proximity to one of the study habitats.**

P10 L31: add "to the" after "compared".

**Changed.**

P12 L2: replace "end-member" with "water mass".

**Changed.**

P12 L9: replace "varience" with "variability".

**Changed.**

P12 L15: replace "end-member" with "water".

**Changed.**

P12 L14-16: please re-write.

**The sentence**

> **"To distinguish ecosystem-driven deviations in the carbonate system from conservative variability, conservative TA and DIC in the off-shore water is estimated and 99 % P.I. are constructed for rTA and rDIC, from offshore observations (Table S2)."**

**has been rephrased to**

> **"The single-end-member conservative mixing model estimates the conservative component of TA and DIC that is inherent from offshore waters. The variability, or error, of this conservative component is estimated from offshore observations by constructing a 99 % P.I. for rTA and rDIC."**

P12 L16: delete one of the "in"'s.

**Changed.**

P12 L20: replace "observations of S" with "the salinity observations".

**Changed.**

P13 L2-3: the statement that DIC is only affected by mixing and metabolic processes is only true if your data is sub-surface data, but if your data is from the surface or close to surface, you also have to consider air-sea gas exchange. This brings me to the fact that there is no information about which depths the samples are collected from. Such information is vital, please include

**As stated in the methods, samples were collected from just above a habitat (ie. The depth of the habitat). Non-habitat samples were collected with Niskin bottles. Nominal depths are recorded in the data spreadsheet that is supplement to the manuscript. We have altered the text to read:**

> **"By comparing relative changes in rTA and rDIC in each habitat, inferences can be made regarding the balance of ecosystem processes within Red Sea coastal habitats (Figure 8, Albright et al., 2013; Challener et al., 2016; Cyronak et al., 2018; Gattuso et al., 1998; Zeebe and Wolf-Gladrow, 2001). If a habitat conforms closely to a linear relationship, it can be inferred that the balance of ecosystem processes is relatively uniform across sites. The slope of the linear relationship indicates the balance of ecosystem processes, with a value determined by the relative proportions of dominant ecosystem processes represented as directional vectors in**

**Figure 8a. Additionally, the intercepts of the linear relationship are inherited from the signals of upstream ecosystems and the amplitude of an observation along this linear relationship is an indication of a combination of metabolic rate and residence time. It also follows that if a habitat doesn't conform closely to a linear relationship, then the balance of ecosystem processes is variable across sites. These inferences cannot be made with other carbonate parameters as, unlike TA and DIC, they are not invariant to changes in T and pressure. Additionally, other carbon variables respond non-linearly to mixing and variations in TA and DIC. Particularly large but linked changes in TA and DIC in the ratio of roughly 1:1 causes other carbonate parameters to change very little (Zeebe and Wolf-Gladrow, 2001). This effect can be observed at the seagrass meadow time series station with the loss of seasonal cycle in other carbonate parameters (Figure 7).**

**As our measurements are made from an open-system, the impacts of air-sea $CO_2$ gas exchange on DIC and the subsequent balance between rTA:rDIC cannot be completely discounted. Air-sea $CO_2$ gas exchange may alter the rTA:rDIC relationship away from the balance of ecosystem processes towards negative rDIC. Offshore waters are $pCO_2$ source and further increases in $pCO_2$ in seagrass meadows and mangrove forests will lead to further $CO_2$ outgassing (Figure 6b; Papaud and Poisson, 1986). However, this effect is relatively small compared to the contribution of ecosystem processes when large ecosystem anomalies are present in seagrass meadows and mangrove forests - a result of high, localised metabolic rates (Burkholz et al. 2019; Ho et al. 2014; Sea et al. 2019). Linked measurements of $CO_2$ gas exchange with metabolic fluxes are required to resolve the magnitude of this effect for Red Sea coastal habitats."**

P21 L10: delete "at is".

**Changed.**

P22 L7-8: delete "and the transition from offshore to coastal waters".

**Changed.**

P25 L4: include the word "which" between "residuals" and "are".

**Changed.**

P26 L4: replace "Forsythe, Malcolm and Moler" with "Forsythe et al."

**Changed.**

P27 L3: remove the first "and".

**Changed.**

Table S1: For Steiner15 data, please write out the years: 2015-2016, and not 2015/6

**Changed.**

Table S3: explain numbers in parenthesis

**Changed.**

Table S3: intercept S of coral reef 3.667, should be 36.67?

**Yes, Changed.**

Table S5: Please rewrite the table text

**Changed.**

**Additional References:**

[revised manuscript text omitted]

**Marked up supplementary material**

**Supplementary Information**

[Figure]

**Figure S1.** Positions of the four time series stations overlaid on a map of the Red Sea produced using © Stamen Design LLC, © Google Maps and R software.

[Figure]

**Figure S2.** Observations of S, T and carbon parameters at four coastal habitats are presented against distance along the south-north central axis (D) on an expanded scale. The circle indicates the location of one outlying mangrove forest observation of TA and DIC that produces un-realistic values of other carbon parameters. Note that observations from time series stations are excluded. Hollow symbols indicate coastal summer observations.

(a)                                      (b)

[Figure]

**Figure S3.** Panel (a) shows a satellite view of the inland-mangrove stand referred to in the text. The picture was obtained using © Google Maps and the red marker indicates the

approximate location the two samples that were taken. Panel (b) shows the rTA and rDIC at this mangrove

forest site (circled red) against other mangrove forests, in an expanded plot of Figure 8e. The red line

indicates the fit of the spatial regression line if this mangrove site is used. All other lines are those displayed

in Figure 8e.

[Figure]

**Figure S4.** Interaction plots as discussed in the text for a) D and habitat type for DIC and b) the seasonal

proxy and habitat type for temperature. The four lines indicate linear trends calculated for offshore water

(OF), transition water (TR), coral reef (CR), seagrass meadow (SM) and mangrove forest (MF).

[Figure]

**Figure S5.** Residual carbon variables observed in the four coastal habitats are presented against distance along the south-north central axis (D). Linear regressions for all combinations of variables are drawn as lines (although none are significant), with associated statistics reported in Table S3. Note that observations from time series stations and the in-land mangroves are excluded. Hollow symbols indicate summer observations.

**Table S1.** Summary of the observations used in this study. Presented is the cruise code, the source of the data, the month and year in which observations were collected and the number of observations that were collected offshore (OS), in transition waters (TR) or at coral reefs (CR), seagrass meadows (SM) and mangrove forests (MF).

| Cruise | Source | Month | Year | OS | TR | CR | SM | MF |
|---|---|---|---|---|---|---|---|---|
| CSM16 | This study | Jan-Apr | 2016 | 13 | 20 | 9 | 11 | 8 |
| CSM17 | This study | Mar | 2017 | 1 | 5 | 11 | 12 | 9 |
| CCF1 | This study | Jan-Mar | 2017 | 0 | 6 | 22 | 5 | 0 |
| CCF2 | This study | Sep-Aug | 2017 | 0 | 2 | 17 | 7 | 0 |
| CRE | This study | May | 2017 | 1 | 0 | 10 | 2 | 2 |
| BPC | This study | April | 2017 | 9 | 6 | 0 | 0 | 0 |
| VOS Pacific Celebes | Hydes et al. (2012) | Nov-Sep | 2007-2009 | 7 | 1 | 0 | 0 | 0 |
| TARA | Picheral et al. (2014) | Jan | 2010 | 3 | 0 | 0 | 0 | 0 |
| WHOI | This study | March | 2010 | 53 | 0 | 0 | 0 | 0 |
| WHOI | This Study | Sep | 2011 | 14 | 0 | 0 | 0 | 0 |
| STEINER15 | Steiner et al. (2018) | Dec-Jan | 2015-2016 | 10 | 0 | 0 | 0 | 0 |
| STEINER18 | Steiner et al. (2018) | March | 2018 | 4 | 1 | 0 | 0 | 0 |
| Total | | | | 115 | 72 | 101 | 69 | 42 |

**Table S2:** Defining statistics for the conservative single-end-member mixing model, as calculated from offshore observations.

| | Residual mean | Residual standard deviation | Lower 99% P.I. bound[+] | Upper 99% P.I. bound[++] | % offshore observations outside the 99% P.I. (excluding/including outliers) |
|---|---|---|---|---|---|
| rTA ($\mu$mol/kg) | 0 | 18.37 | -47.31 | 47.31 | 1.0/4.3 |
| rDIC ($\mu$mol/kg) | -0.66 | 24.59 | -64.00 | 62.68 | 1.9/4.3 |
| rpH | $-3 \times 10^{-4}$ | $2.61 \times 10^{-2}$ | $-6.76 \times 10^{-2}$ | $6.70 \times 10^{-2}$ | 3.8/6.1 |
| r$p$CO$_2$ ($\mu$atm) | -0.23 | 29.68 | -76.70 | 76.24 | 3.8/7.0 |
| r$\Omega_{Ar}$ | 0.0006 | 0.1816 | -0.4672 | 0.4684 | 4.8/7.8 |

[+] Residual mean – 2.576*Residual standard deviation
[++] Residual mean + 2.576*Residual standard deviation

**Table S3.** Descriptive statistics for regressions of different variables against D constructed with observations from offshore waters and the four habitat types. Note that observations from time series stations are not included. Slope (±SD), intercept (±SD), Pearson's correlation coefficient ($r^2$), the test statistic (F) and p-value (p) are reported for each individual test.

**Table S4.** Descriptive statistics for carbon variables for coastal observations. Note that observations from time series stations are excluded. The number of observations (n), mean, median, standard deviation, maximum values (max) and minimum value (min) are presented for each habitat group and variable combination.

**Table S5**. Results of one-way WR-ANOVA and corresponding boot-strapped post-hoc t-tests for differences in medians between habitat groups; offshore waters (OS), transition waters (TR), coral reefs (CR), seagrass meadows (SM) and mangrove forests (MF). Note that observations from time series stations are excluded. Tests statistics (F for WR-ANOVA and Ψ_hat for post-hoc) and p-values (p) are reported for each individual test.

**Table S6.** Statistics for regressions of different variables against the seasonal proxy (SP) at the four time series stations. Pearson's correlation coefficient ($r^2$), the test statistic (F) and p-value (p) are reported for each individual test.

**Table S7.** Descriptive statistics for studied variables at the four time series stations. The number of observations (n), mean, median, standard deviation, maximum values (max) and minimum value (min) are presented for each time series station and variable combination.

**Table S8**. Results from one-way WR-ANOVA and corresponding boot-strapped post-hoc t-tests for differences in medians between the four time series stations; transition water (TR), coral reef (CR), seagrass meadow (SM) and mangrove forest (MF). Tests statistics (F for WR-ANOVA and Ψ_hat for post-hoc) and p-values (p) are reported for each individual test.

**Supplementary R Code: WRS2_post_hoc.R**

This code contains 1) the med1way.crit function which is an internal function of the WRS2 package

(source: https://github.com/cran/WRS2/blob/master/R/med1way.crit.R) and 2) an adapted version of the

mcppb20 function, which is contained in the WRS2 package (source:

https://github.com/cran/WRS2/blob/master/R/mcppb20.R), that performs bootstrapped t-tests for differences

in medians.

---

## Author Response (AR3)

**Authors Response**

Anomalies in the Carbonate System of Red Sea Coastal Habitats
Kimberlee Baldry, Vincent Saderne, Daniel C. McCorkle, James H. Churchill, Susana Agusti and
Carlos M. Duarte

We thank the editors for inviting us to resubmit our article, and the reviewers who took the time to read and comment on it. Within this document we have included:

**List of relevant changes**

- Changed the acknowledgement section to refer to only one anonymous reviewer, not two.

**Point-by -point response to Zvi Steiner**

**Author response to the editor**

The authors have responded in **bold.**

Please just consider a minor technical corrections: your manuscript was reviewed twice by the same anonymous reviewer, not by two anonymous reviewers as you stated in the acknowledgements. Please correct that point.

**The last sentence of the acknowledgements now reads:**

[revised manuscript text omitted]